# A Study of the Deformation Law of the Surrounding Rock of a Laminated Roadway Based on FLAC3D Secondary Development

Tuo Wang [1,2], Jucai Chang [1,2] and Hongda Wang [1,2,*]

1   State Key Laboratory of Mining Response and Disaster Prevention and Control in Deep Coal Mines, Anhui University of Science and Technology, Huainan 232001, China; twang1089@126.com (T.W.); jcchang@aust.edu.cn (J.C.)
2   School of Mining Engineering, Anhui University of Science and Technology, Huainan 232001, China
*   Correspondence: aust_whd@163.com

**Abstract:** To investigate and analyze the influence of different stress environments on the deformation and destabilization of the rocks surrounding laminated roadways under high stress, this study conducted numerical simulations of coal–rock combination under different circumferential pressures and of the surrounding rocks of highly stressed laminated roadways under different lateral pressure coefficients. In addition, a new custom constitutive structure model was constructed based on the Mohr–Coulomb criterion and realized in FLAC3D software by combining field working conditions. The model was then developed in FLAC3D software for a second time. The results show that the calculated results of the model in this study are in good agreement with the experimental results and the errors are small, while the calculated results of the Mohr–Coulomb model differ from the experimental values under two types of surrounding rock pressure. The deformation of the Mohr–Coulomb model is significantly smaller than that of the customized model, which verifies the reasonableness and superiority of the self-built model in combination with the field conditions. This provides theoretical and practical bases for the design and optimization of stratigraphic roadway support in underground coal mines.

**Keywords:** stress environment; laminated roadway envelope; Mohr–Coulomb criterion; FLAC3D numerical simulation secondary development; damage constitutive model

## 1. Introduction

In recent years, coal mining has increased in depth and intensity. The high-stress laminated roadway envelope presented by coal mining gradually develops to a deeper part, making the original roadway support scheme based on the complete rock formation inadequate [1–5]. Therefore, there is an urgent need to explore a new support theory suitable for laminated roadway envelopes under high-stress conditions. Firstly, it is essential to study the influence of different stress environments on the support of deep ground engineering roadway rock enclosures containing laminated roadways.

In actual geological engineering, there are a large number of fractured rocks. The original cracks and pores inside the rock are closed during the initial loading process, exhibiting a characteristic of volume compression. Strength criterion and the constitutive structure relationship are important components of the rock mechanics theory that reflect the strength characteristics and deformation properties of rocks, respectively, and are the basis for the theoretical analysis and numerical simulation studies of rock mechanics [6–11]. Further, the mechanical testing of a coal rock body can not only provide the basic mechanical parameters of the rock, but also has an important reference significance in revealing the fracture evolution and failure mechanism of the roadway enclosure. Many scholars have carried out mechanical property tests on coal and rock bodies, including conventional triaxial tests

on pure coal and pure rock bodies, and true triaxial tests and mechanical property tests on rocks under different stress paths. Meanwhile, some experts and scholars have also conducted mechanical tests and revealed the strength characteristics and damage forms of coal–rock combinations [12]. Shallow rocks mostly exhibit linear brittle characteristics, and their strength theory is generally based on the linear Mohr–Coulomb criterion, while deep rocks are in a special environment of high ground stress, and their deformation and damage characteristics show obvious nonlinearity [13–15].

Liu [16] conducted conventional triaxial mechanical properties tests of coal and rock through the MTS815 test system to study the deformation, strength parameters and damage characteristics of coal rock. Zhang [17] numerically simulated the cracking process of rock-like materials containing a single crack under uniaxial vertical compression using a parallel bonding model based on a particle flow numerical modeling approach. Chen [18,19] established the process characteristics, the damage evolution law and the Coulomb criterion for the strength of rock micro-elements, and numerically simulated the damage evolution process of rocks under different peritectic pressures by using RFPA2D (two-dimensional rock failure process analysis) numerical simulation software. Song et al. [20] conducted a study of the whole process of triaxial compression damage and carried out post-peak cyclic loading and unloading tests on rock samples to explore the deformation and damage characteristics of the surrounding rock after exceeding the peak strength. Liu et al. [21] analyzed energy evolution law in the process of coal body damage via uniaxial compression testing, and discerned the precursor characteristics of coal body destabilization damage. Zhou et al. [22] analyzed the physical state, damage mode and propensity of the coal during protrusion based on the results of large protrusion physical simulation tests. Wen [23] conducted uniaxial compression tests on lignite specimens with different crack angles under laboratory conditions to investigate the role of primary fractures in the coal body on the potential failure mechanism.

However, for roadways containing weakly laminated faces whose overburden loads are carried by both coal and rock, the safety and stability of the roadway are jointly determined by the roof rock and the coal body. The deformation variability of rock and coal is one of the important reasons for the uneven large deformation in roadways. Therefore, study of the deformation, strength characteristics and failure characteristics of a single coal or rock body cannot accurately reveal the force characteristics of such a roadway, nor can it directly indicate the cause of uneven large deformation in the roadway.

Zuo [24–26] studied the mechanical characteristics of a coal–rock combination with the same size of φ 35 mm × 35 mm experimentally, using the surrounding rock pressure as the variable. Yang [27] made a coal–rock three–body combination specimen and combined it with PFC2D numerical simulation software to study the mechanical properties and crack expansion of coal–rock combination under uniaxial compression. Guo et al. [28] conducted a study to address the key concerns of experts and scholars in the field on the butterfly damage theory, and analyzed the reliability of theoretical calculations and the applicability of the butterfly damage theory under different roadway section shapes and laminated surrounding rock conditions through theoretical analysis and numerical simulation. Wang [29,30] analyzed the generation and formation process of plastic zone in the surrounding rock of high-stress soft rock roadway by means of theoretical analysis, numerical calculation and field test. Yuan et al. [31] established a mechanical model of a circular roadway under deep dynamic pressure, and derived the implicit equation of the plastic zone boundary for the large deformation and destabilization damage of a deep dynamic pressure back mining roadway and its control problems. Li et al. [32] elucidated the deformation and damage characteristics of a roadway enclosure under various complex conditions to ensure the stability of a down-hole roadway in the comprehensive release of a thick coal seam containing gangue and to ensure the safe and efficient mining of the working face. Pei et al. [33] used numerical simulation to analyze the stress distribution, plastic zone distribution and displacement of the rocks surrounding a down-hole roadway under different gangue conditions. Dai [34] carried out rock mass classification and a rock mechanical pa-

rameter estimation based on field geological investigation and rock mechanics experiments. Based on the mine mining plan, they established a numerical model to analyze the stress, plastic zone and displacement change law of the rocks surrounding the roadway under the influence of mining. For the modeling of coal rock interfaces, Sun [35] studied the mechanical characteristics (plastic zone, stress, and displacement) of a coal–rock composite structure under different interface connection modes by numerical simulation, and analyzed the energy by FISH language in FLAC3D. Zhao [36] proposed a universal method to achieve the pre-perception and accurate recognition of coal–rock interfaces; meanwhile, a convolutional block attention module was employed to improve the coal–rock interface identification ability of the proposed network. Zuo [37] presented experimental and numerical investigations on the response of a rock–coal–rock bimaterial composite structure under triaxial compression and the corresponding numerical simulation were carried out by using the particle flow code.

In practical engineering, due to the contact between the roof rock and the coal seam to be mined, thus forming a laminated roadway enclosure under high stress and different lateral pressure coefficients, the mechanical characteristics of a high-stress laminated roadway enclosure are different. Therefore, this paper carries out a study of the effect of different stress environments on the stability of high-stress laminated roadway enclosures. The innovations are as follows: the stability of a high-stress, layered roadway surrounding rock under different pressure measurement coefficients was investigated based on the secondary development and application of FLAC3D. This is important for further promoting the study of the destabilization law of laminated roadways under high-stress conditions.

## 2. Ontological Modeling of Damage in Coal–Rock Combination

On the basis of the Mohr–Coulomb criterion, the micro-element strength measurement method proposed by Wang [38] that can consider the damage threshold introduces the derivation method and results of the damage model of coal in the coal–rock combination. The model fits well with the test data. Therefore, this article deduces its three-dimensional difference form and applies it to the secondary development of the FLAC3D numerical calculation model for the analysis of tunnel excavation.

The theoretical model derived is Equation (1):

$$\sigma_{c1} = (1 - D)E_c\varepsilon_{c1} + 2(1 - D)\mu_c\sigma_{c3} + DQ \tag{1}$$

Equation (1) is the damage constitutive model of coal in the coal–rock combination under conventional triaxial loading.

where $Q = k\tan^2\alpha + 2c\tan\alpha$, $E_c$ and $\varepsilon_{c1}$ are the elastic modulus and axial strain of coal, respectively, and $\mu_c$ stands for the Poisson's ratio of coal. $D$ represents the damage variable, which is the percentage of cracks or voids in the entire material volume.

The relationship between the total stress tensor $\sigma_{ij}$, and the spherical stress tensor $\sigma_m$ and the partial stress tensor $S_{ij}$ using the central difference method can be expressed as Equation (2):

$$\sigma_{ij} = S_{ij} + \sigma_m\delta_{ij} \tag{2}$$

where $\delta_{ij}$ denotes the Kronecker delta symbol.

In the elastic phase, the ratio of the stress bias increment to the strain bias increment is a constant 2G, and $G$ is the shear modulus. Its three–dimensional tensor form is Equation (3):

$$S_{ij} = 2Ge_{ij} \tag{3}$$

The relationship between the stress increment and strain increment within time step $\Delta t$ is:

$$\Delta\sigma_{ij} = 2G\Delta e_{ij} + K\Delta\varepsilon_{kk}\delta_{ij} \tag{4}$$

where $S_{ij}$ denotes the stress bias; $e_{ij}$ represents the strain bias; $\Delta e_{ij}$ stands for the bias strain increment; $\sigma_{ij}$ is the stress tensor at moment $t$; and $\Delta\sigma_{ij}$ denotes the stress tensor increment at moment $t$; $K$ is the bulk modulus; $G$ is the shear modulus.

The relationship between the stress increment and the strain increment of the damaged body within time step $\Delta t$ can be rewritten as Equation (5):

$$\Delta\sigma_{ij} = 2(1 - D_t)G\Delta e_{ij} + K(1 - D_t)\Delta\varepsilon_{kk}\delta_{ij} + D_t Q_t \tag{5}$$

Denote the new partial stress state within a time increment step as new, i.e., the stress at step $i$, and the old partial stress state within a time step as old, i.e., the stress at step $i - 1$. Therefore,

$$\overline{S}_{ij} = \frac{S_{ij}^N + S_{ij}^O}{2} \tag{6}$$

$$\overline{e}_{ij} = \frac{e_{ij}^N + e_{ij}^O}{2} \tag{7}$$

$$\Delta e_{ij} = e_{ij}^N - e_{ij}^O \tag{8}$$

where $S_{ij}$, $S_{ij}^N$ and $S_{ij}^O$ denote the average bias stress and the new and old bias stresses in one time increment step, respectively. $e_{ij}$, $e_{ij}^N$ and $e_{ij}^O$ denote the average bias strain and the new and old bias strains within one time increment step, respectively.

Then, according to the above equations, the differential form of Equation (1) can be rewritten as Equations (9) and (10):

$$\sigma_m^N = \frac{1}{3}\sigma_{kk}^O + A\Delta\varepsilon_{kk}\delta_{ij} \tag{9}$$

$$S_{ij}^N = S_{ij}^O + B\Delta e_{ij} \tag{10}$$

where $A$ and $B$ in the above equation are, respectively.

$$A = \frac{GKE(1 - D_t)}{6KE + 2GED_t - 3KGD_t(1 + 2k_{pt}) - 9GK} \tag{11}$$

$$B = \frac{3KGE(1 - D_t)}{GE + 3KED_t + 3KGD_t(k_{pt} - 1)} \tag{12}$$

where $D_t$ is the damage variable.

### 2.1. Writing and Loading of FLAC3D Numerical Simulations of Custom Constitutive Models

The computational flow of the custom constitutive structure model is:

1. Assign initial values to the state variables of the model cell;

2. Calculate the new partial stress, partial strain and total stress and total strain of the cell based on the strain increment, based on the custom constitutive increment equation;

3. Determine whether the model reaches the damage state, and update the damage variable if it reaches the damage state;

4. Determine whether the model reaches the yield state, and if it reaches the yield state, determine its damage state, whether it is tensile damage or shear damage;

5. In the case of shear damage, the stress is corrected using the uncorrelated flow rule; in the case of tensile damage, the stress is corrected using the correlated flow rule;

6. Calculation of unbalanced forces, and rates and displacements of nodes using the new stresses after correction. In the FLAC3D main program, the FISH programming language was used to judge whether the unbalanced force converges or not. If not, return

to the second step using the updated damage variables, otherwise, the program ends. The program is run and loaded as shown in Figure 1.

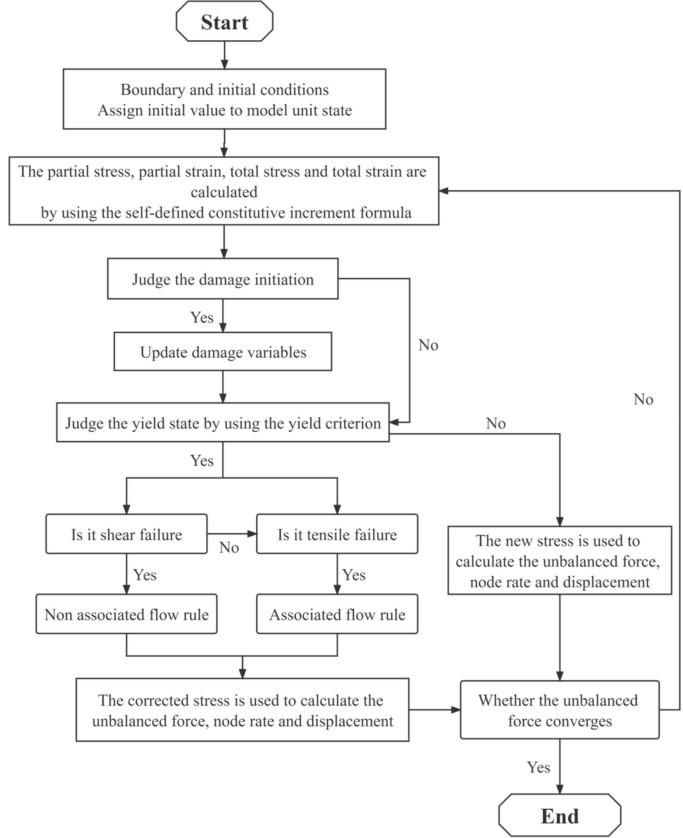

**Figure 1.** Custom constitutive model FLAC3D running program.

## 2.2. Numerical Simulation of Coal–Rock Combination with Different Surrounding Pressures

After loading the constitutive model, verify its calculation results and experimental results. The height and diameter of the rock and coal established by numerical simulation are both 50 mm, as shown in Figure 2. The loaded surrounding rock pressures are 5 MPa and 10 MPa, respectively, and the axial the loading velocity is $1 \times 10^{-7}$ mm/step. The parameters F0, and m are based on Table 1. The physical and mechanical parameters of the rock selected for the simulation test are consistent with the test, as shown in Table 2.

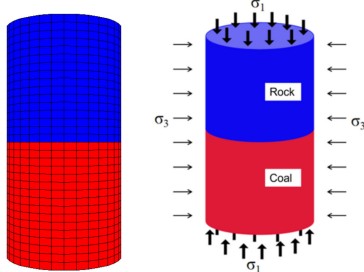

**Figure 2.** Conventional triaxial numerical calculation model and loading. The arrow in the figure represents the direction of stress action.

**Table 1.** Calculated values of damage nonlinear model parameters.

| Surrounding Rock Pressure σ | δ | Micro-Element Intensity Weibull Distribution Parameters $F_0$ | Micro-Element Intensity Weibull Distribution Parameters m | Goodness of Fit $R^2$ |
|---|---|---|---|---|
| 5 MPa | 0.33 | 14.7538 | 3.2466 | 0.944 |
| | 0.5 | 13.5508 | 5.9780 | 0.906 |
| | 0.67 | 14.7134 | 3.9261 | 0.946 |
| | 0.75 | 12.7565 | 4.9362 | 0.961 |
| 10 MPa | 0.33 | 21.9997 | 5.8555 | 0.901 |
| | 0.5 | 25.6579 | 3.8349 | 0.893 |
| | 0.67 | 27.1918 | 4.5514 | 0.875 |
| | 0.75 | 20.4382 | 3.8718 | 0.853 |

**Table 2.** Physical and mechanical parameters of rock.

| Combined Rock Mass | E/GPa | v | φ/° | c/MPa |
|---|---|---|---|---|
| Coal | 1.22 | 0.33 | 8 | 1.43 |
| Rock (sandstone) | 6.88 | 0.28 | 35 | 2.52 |

In view of the space to insert a figure as an example, Figure 3 shows the stress distribution cloud and the stress–strain curve of the coal–rock combination when the surrounding pressure is 10 MPa, while Figure 4 shows the fitting results of the custom principal structure testing under different surrounding pressure conditions.

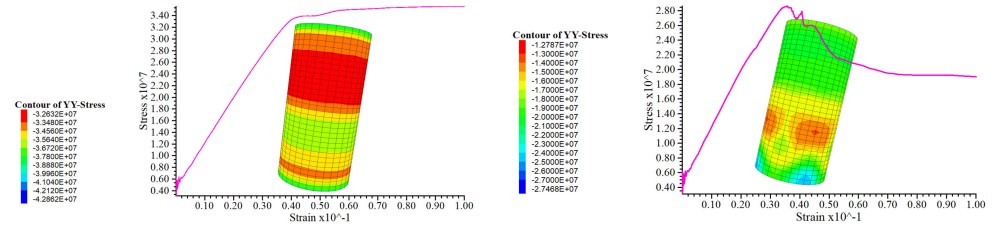

**Figure 3.** Deformation and stress–strain trend of the model under 10 MPa surrounding rock pressure.

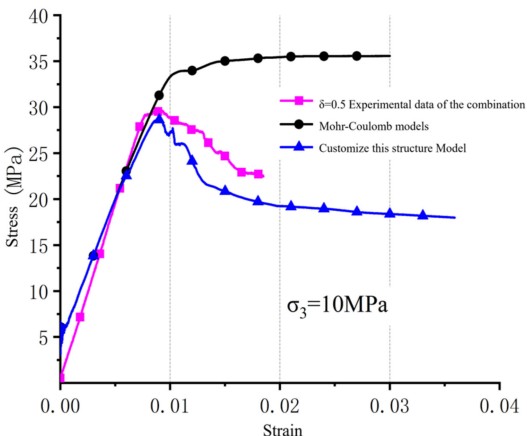

**Figure 4.** Fitting of the customization constitutive model and the Mohr–Coulomb and test curve with 10 MPa of surrounding rock pressure (experimental data cited from Wang [38]).

As shown in Figure 3, it can be seen that the stress distribution of the coal–rock combination using the Mohr–Coulomb constitutive model is relatively uniform, while the stresses of the custom constitutive model clearly tend to decrease in subregions, and there are

obvious low-stress regions in the coal body. This is because the stress in the custom constitutive model decays gradually with loading after the material reaches yield, which is also reflected in its stress and strain curves.

It can be seen in Figure 4 that the coal–rock combination is kept in a linear elastic state initially with a constant stress–strain ratio when the Mohr–Coulomb constitutive simulation is used. The stress remains constant and the strain keeps increasing when the plastic state is reached, as the Mohr–Coulomb is an ideal elastic–plastic model that maintains ideal plastic flow characteristics after the peak stress is reached, which differs greatly from the actual stress–strain relationship of the rock.

Moreover, the peak stresses and strains of the coal–rock combination calculated with the Mohr–Coulomb model were 17.98 MPa, 0.006 and 33.97 MPa, 0.012 for the two surrounding rock pressures, respectively; while the peak stresses and strains of the custom constitutive model were 15.82 MPa, 0.00507 and 28.73 MPa, 0.00895, respectively. The difference between the calculated results of the Mohr–Coulomb model and the experimental values is large, with a maximum difference of 13.5% in peak stress and 15.05% in peak strain, while the maximum differences between the corresponding peak stress and strain of the custom constitutive model are 4.3% and 3.7%.

Therefore, it can be seen that the custom constitutive model can better fit and respond to the stress and strain development of the rock, and can better reflect the stress and deformation of the rock surrounding the underground space compared with the traditional Mohr–Coulomb constitutive model.

## 3. Influence of Lateral Pressure Coefficient on the Stress and Deformation Pattern of Laminated Roadways

### 3.1. Analog Parameter Selection

In the process of roadway layout, in reality, the roadway is often in a three-way unequal pressure state due to the influence of mining stress and the increases in burial depth. When excavating tunnels or mining coal seams, the horizontal stress on the goaf side sharply decreases, and the lateral pressure coefficient of the surrounding rock is relatively small. The horizontal stress variation in the non-goaf side surrounding rock is relatively small. Due to the generated stress concentration and large vertical stress addition, the lateral pressure coefficient of the surrounding rock can change from 0 to 2, or even greater.

Therefore, in this section, the simulation process is divided into two cases: the first group is to keep the vertical pressure of the roadway constant at 10 MPa, with the horizontal pressure varying between 1 MPa and 30 MPa; and the second group is to keep the horizontal pressure of the roadway constant at 10 MPa, with the vertical pressure varying between 1 MPa and 30 MPa, thus changing the lateral pressure coefficient of the surrounding rock, as shown in Figures 5 and 6.

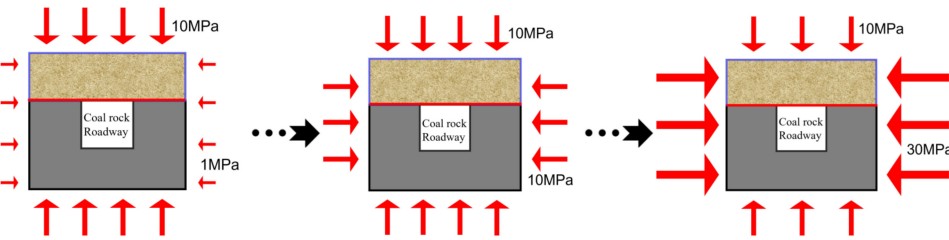

**Figure 5.** The first group of surrounding rock pressure loading methods. The arrow in the figure represents the direction of stress action.

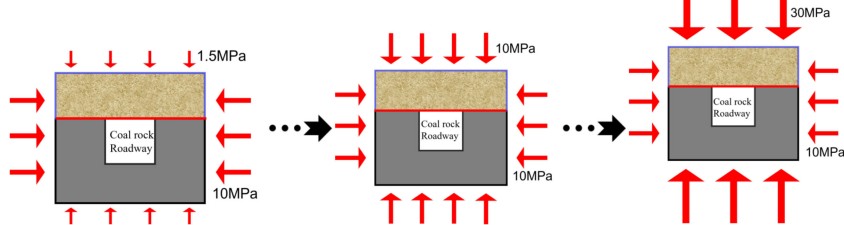

**Figure 6.** The second group of surrounding rock pressure loading methods. The arrow in the figure represents the direction of stress action.

The simulated horizontal and vertical pressure values are shown in Tables 3 and 4.

**Table 3.** The first group of vertical pressure remains unchanged, while the horizontal force changes.

| Lateral pressure coefficient | 0.1 | 0.15 | 0.2 | 0.4 | 0.6 | 1 | 1.4 | 1.8 | 2.2 | 2.6 | 2.8 |
|---|---|---|---|---|---|---|---|---|---|---|---|
| Horizontal force change/MPa | 1 | 1.5 | 2 | 4 | 6 | 10 | 14 | 18 | 22 | 26 | 28 |
| Constant vertical force/MPa | | | | | | 10 | | | | | |

**Table 4.** The second group of vertical pressure changes, while the horizontal force remains unchanged.

| Lateral pressure coefficient | 0.333 | 0.384 | 0.555 | 0.714 | 1 | 1.67 | 2.5 | 6.7 |
|---|---|---|---|---|---|---|---|---|
| Constant horizontal stress/MPa | | | | | 10 | | | |
| Vertical stress change/MPa | 30 | 26 | 18 | 14 | 10 | 6 | 4 | 1.5 |

### 3.2. Roadway Surrounding Rock Stress Distribution Law

Figures 7 and 8 show the horizontal stress distribution in the laminated roadway under two different sets of loading methods, respectively, and it can be seen that:

(1) Under the Mohr–Coulomb constitutive model, when the lateral pressure coefficient is less than 1, the horizontal stress is concentrated at the coal–rock interface, and the peak point at the interface is 2.5~3 m away from the top corner of the roadway.

(2) As the horizontal stress rises, i.e., the lateral pressure coefficient of the surrounding rock rises, the stress concentration area gradually shifts to the floor.

(3) The custom model has a stress concentration area farther from the roadway and a larger pressure relief zone around the roadway than the Mohr–Coulomb constitutive model.

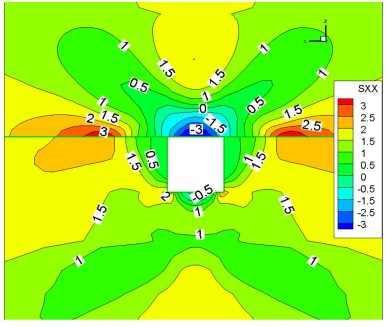 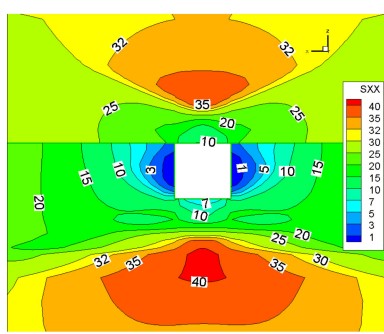

Lateral pressure coefficient 0.15 Lateral pressure coefficient 3.0

**Figure 7.** The first group of horizontal stress distribution with the Mohr–Coulomb model. The line in figure is stress contour line.

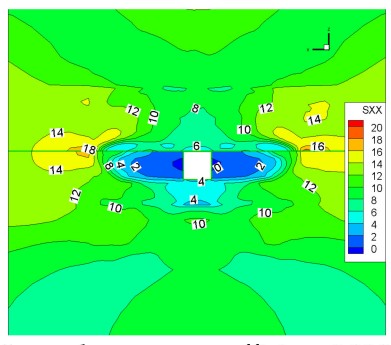 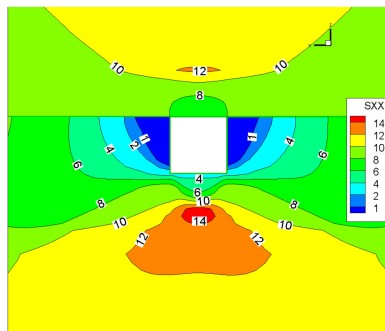

Lateral pressure coefficient 0.333 Lateral pressure coefficient 6.7

**Figure 8.** The second group of horizontal stress distribution with customized constitutive model. The line in figure is stress contour line. The line in figure is stress contour line.

It can be seen in Figure 8 that:

(1) Under the two constitutive models, when the lateral pressure coefficient is less than 1, the horizontal stress is concentrated at the coal–rock interface, and as the initial horizontal stress increases, i.e., the lateral pressure coefficient of the surrounding rock increases, the stress concentration area gradually shifts to the floor;

(2) The distance between the stress concentration area and the roadway of the custom constitutive model is greater than that of the Mohr–Coulomb constitutive model, and the stress value is slightly greater than that of the Mohr–Coulomb constitutive model.

Figure 9 shows the vertical stress distribution of the tunnel surrounding rock under different lateral pressure coefficients when changing the horizontal stress, which can be seen as follows:

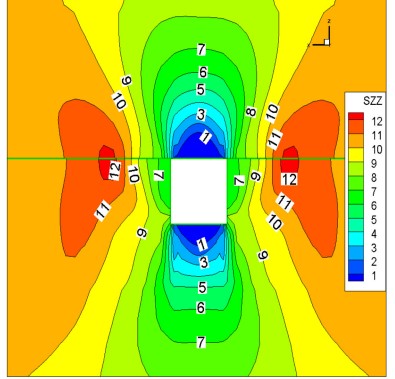 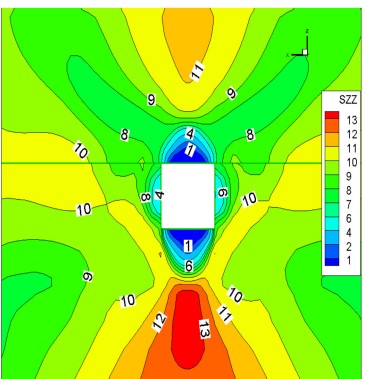

Lateral pressure coefficient 0.15      Lateral pressure coefficient 3.0

**Figure 9.** Vertical stress distribution of the surrounding rock of the first group of roadways under the Mohr–Coulomb principal structure model. The line in figure is stress contour line.

(1) Under the Mohr–Coulomb principal model, when the lateral pressure coefficient is small, the vertical stress concentration area is distributed near the roadway interface, and when the lateral pressure coefficient is 0.15, the peak vertical stress is 12.4 MPa, which is 2.5~3 m away from the top corner of the roadway;

(2) When the lateral pressure coefficient is 3, the peak vertical stress is 13.7 MPa, and the peak concentration area is shifted to the floor of the roadway. At 2.8 m, the stress value above the top plate also increases gradually;

(3) With the increase in the lateral pressure coefficient, the peak vertical stress does not increase much, while the vertical stress around the roadway gradually decreases and the stress reduction area gradually increases;

(4) Under the custom constitutive model, the vertical stress distribution trend is the same as that of the Mohr–Coulomb constitutive model;

(5) As the lateral pressure coefficient increases, the stress concentration area is gradually shifted from the roadway sidewall to the floor and top of the roadway, the vertical stress concentration area and peak value of the floor are slightly larger than that of the top plate and are gradually moved away from the roadway, and the stress relief area around the roadway is gradually increased.

Figure 10 shows the vertical stress distribution in the surrounding rock of the second group of roadways, which can be seen as follows:

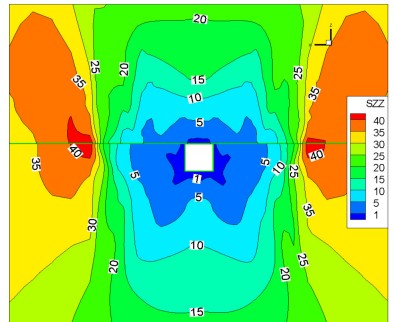 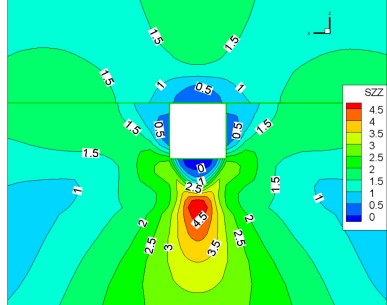

Lateral pressure coefficient 0.333      Lateral pressure coefficient 6.7

**Figure 10.** Vertical stress distribution of the surrounding rock in the second group of roadways under customized constitutive model. The line in figure is stress contour line.

(1)　Under the two constitutive models, the vertical stress is concentrated at the coal–rock interface when the lateral pressure coefficient is less than 1. As the initial vertical stress decreases, i.e., the lateral pressure coefficient of the surrounding rock increases, the stress concentration area shifts to the floor when the initial vertical stress is less than the horizontal stress;

(2)　The distance between the vertical stress concentration area and the roadway is greater in the custom model than in the Mohr–Coulomb model, and the stress value is slightly greater than in the Mohr–Coulomb model. For example, when the lateral pressure coefficients are 0.333 and 6.7, the maximum stress values of the customized constitutive model reach 43.4 MPa and 4.9 MPa, while the Mohr–Coulomb constitutive model is 39.6 MPa and 3.6 MPa.

### 3.3. Deformation Law of Roadway Surrounding Rock

Figure 11a–c show the variations in the maximum displacement of the sidewall, floor and top of the tunnel surrounding rock under different lateral pressure coefficients when the vertical stress is constant, respectively, and Figure 11d shows the enlargement of the shaded areas in regions A, B and C.

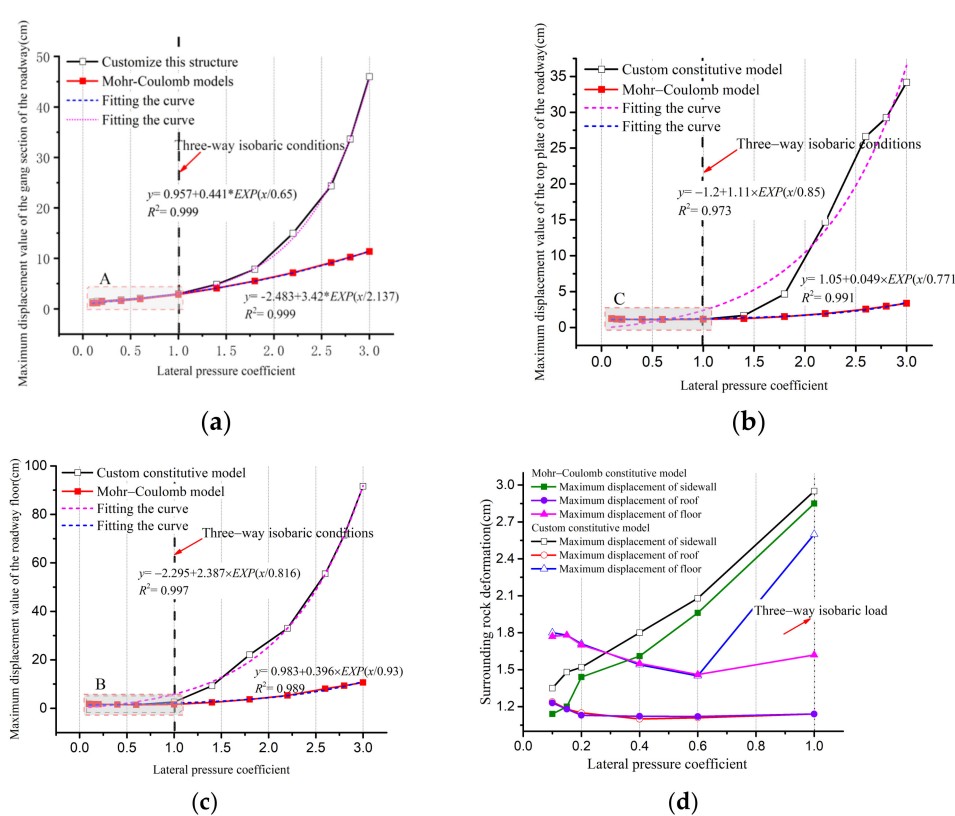

**Figure 11.** The influence of the first set of lateral pressure coefficients on the maximum displacement value of the surrounding rock of the roadway. (**a**) maximum displacement value of the sidewall of the roadway; (**b**) maximum displacement value of the floor; (**c**) maximum displacement value of the roof; (**d**) regions A, B and C.

The figures show that:

(1)　The maximum displacement of the tunnel surrounding rock increases exponentially with the increase in the lateral pressure coefficient;

(2)　The deformation of the surrounding rock in the roadway is small, when the lateral pressure coefficient is less than 1. As the lateral pressure coefficient increases, the increase in deformation and the difference of deformation between the two constitutive models is small;

(3) When the lateral pressure coefficient is greater than 1, the growth rate of surrounding rock deformation accelerates, and the difference between the two constitutive models gradually increases;

(4) When the lateral pressure coefficient is 3, the maximum displacements of the custom constitutive model sidewall, roof and floor are 46 cm, 34.17 cm and 91.56 cm, respectively, and the corresponding maximum displacements of the Mohr–Coulomb constitutive model are 11.37 cm, 3.38 cm and 10.65 cm, respectively, with the calculation results of the two constitutive models differing greatly;

(5) When the lateral pressure coefficient is 3, the maximum displacements of the custom constitutive model sidewall, roof and floor are 22.1 times, 30.8 times and 63.1 times of the lateral pressure coefficient of 0.6, respectively, and the corresponding Mohr–Coulomb constitutive models are 5.8 times, 3.0 times and 7.3 times, respectively.

The maximum displacement trend of the surrounding rock with the lateral pressure coefficient is in exponential form, with R2 ranging between 0.97 and 0.999, which is a good fit. The specific fitting statistics are shown in Table 5.

**Table 5.** The fitting function of the maximum displacement value of the first group of surrounding rocks.

| Principal Structure Model | Location of Roadway Enclosure | Fitting Function | $R^2$ |
|---|---|---|---|
| Mohr–Coulomb | sidewall | $y = -2.483 + 3.42 \times e^x/2.137$ | 0.999 |
| | Roof | $y = 1.05 + 0.049 \times e^x/0.771$ | 0.991 |
| | Floor | $y = 0.983 + 0.396 \times e^x/0.927$ | 0.989 |
| Customization | sidewall | $y = 0.957 + 0.441 \times e^x/0.65$ | 0.999 |
| | Roof | $y = -1.2 + 1.11 \times e^x/0.85$ | 0.973 |
| | Floor | $y = -2.295 + 2.387 \times e^x/0.816$ | 0.997 |

Figure 12a–c show the changes in the maximum displacement of the sidewall, floor and roof of the roadway surrounding rocks under different lateral pressure coefficients when the horizontal stress is constant, respectively, and Figure 12d shows the enlargement of the shaded areas in (b) and (c). It can be seen in the figure that:

(1) As the lateral pressure coefficient increases, the maximum displacement values of the surrounding rocks of the sidewall, roof and floor of the tunnel under the two ontogenetic models decrease exponentially. The deformation of the Mohr–Coulomb model is significantly smaller than that of the custom model, with 9.73 cm, 5.0 cm and 6.7 cm, respectively;

(2) The trend of the maximum displacement of the surrounding rock with the lateral pressure coefficient conforms to the exponential form, fitted with the exponential function, R2 ranging between 0.957 and 0.998, which is a good fit. The specific fitting function statistics are shown in Table 6.

**Table 6.** The fitting function of the maximum displacement value of the second group of surrounding rocks.

| Principal Structure Model | Location of Roadway Enclosure | Fitting Function | $R^2$ |
|---|---|---|---|
| Mohr–Coulomb | Sidewall | $y = 2.94 + 53.27 \times e-x/0.16$ | 0.994 |
| | Roof | $y = 0.52 + 15.48 \times e-x/0.27$ | 0.957 |
| | Floor | $y = 0.856 + 19.75 \times e-x/0.254$ | 0.96 |
| Customization | Sidewall | $y = 3.61 + 1711 \times e-x/0.1$ | 0.998 |
| | Roof | $y = 1.03 + 155.97 \times e-x/0.117$ | 0.995 |
| | Floor | $y = 1.86 + 133.29 \times e-x/0.123$ | 0.983 |

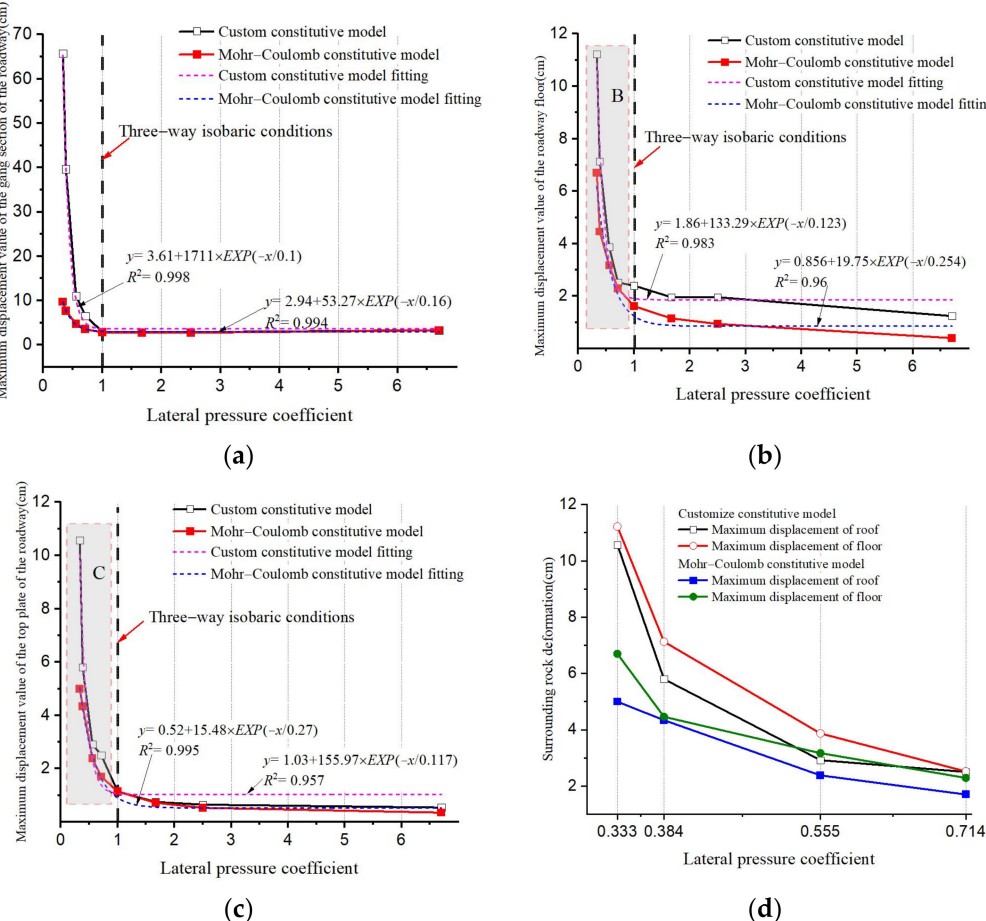

**Figure 12.** The influence of the second set of lateral pressure coefficients on the maximum displacement value of the surrounding rock of the roadway. (**a**) Maximum displacement value of the surrounding rock at the sidewall of the roadway; (**b**) maximum displacement value of the floor surrounding rock; (**c**) maximum displacement value of the roof surrounding rock; (**d**) regions B and C.

### 3.4. The Distribution Pattern of the Plastic Zone in the Roadway Surrounding Rock

Comparison and extraction of the distribution pattern of the plastic zone in the roadway surrounding rock under different lateral pressure coefficients using the Mohr–Coulomb and custom constitutive models.

Figure 13 shows the distribution of the plastic zone of the roadway surrounding rock under a partial pressure measurement coefficient, which can be seen:

(1) The plastic zone of the surrounding rock tends to decrease and then increase as the lateral pressure coefficient increases. When the lateral pressure coefficient is less than 1, the plastic zone gradually decreases; when the lateral pressure coefficient is greater than 1, the plastic zone gradually increases and the expansion rate increases faster;

(2) Under the Mohr–Coulomb model, when the lateral pressure coefficient is less than 0.4 or more than 2.2, the plastic zone of the roadway will show "butterfly" damage, and the plastic zone of the two top and two floor corners will be more developed, while the plastic zone of the floor corner is larger than that of the top corner;

(3) Under the custom constitutive model, when the lateral pressure coefficient is less than 0.4, the roadway shows "butterfly" damage; when the lateral pressure coefficient is greater than 1.0, with the increase in the lateral pressure coefficient, the expansion of the plastic zone of the roadway surrounding rock is accelerated, and the plastic zone of the roadway appears similar to "butterfly" damage, with the plastic zone of the two top corners and the two floor corners developing more than the sidewall and middle-top of the roadway;

(4)　Under the same lateral pressure coefficient, the plastic zone of the surrounding rock in the Mohr–Coulomb model is smaller than that in the custom constitutive model, and the difference is more obvious when the lateral pressure coefficient is greater than 1. The maximum extension of the plastic zone of the Mohr–Coulomb model is 12 m, 4.2 m and 4.25 m for the sidewall, roof and floor, respectively, and the maximum extension of the plastic zone of the corresponding custom constitutive model is 16 m, 10 m and 11 m, respectively, which is much larger than that of the Mohr–Coulomb model.

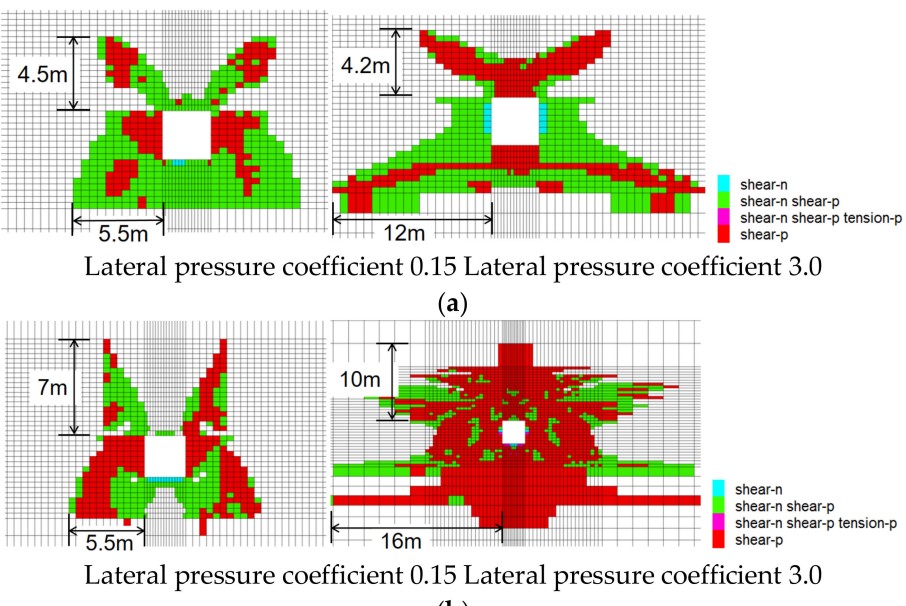

**Figure 13.** The plastic zone distribution of the surrounding rock in the first group of model roadways. (**a**) Plastic zone of the Mohr–Coulomb constitutive model roadway perimeter rock. (**b**) Customized constitutive model roadway perimeter rock plasticity zone.

Figure 14 shows the development of the plastic zone of the surrounding rock under different lateral pressure coefficients of the second group of tunnels, which can be seen as follows:

(1)　When the lateral pressure coefficient is small, the roadway is a "butterfly– shaped" development, the top plate plastic zone is smaller than the floor and the roadway two "shoulder angle" plastic zone depth is greater than the depth of the two floor angles;

(2)　Under the same lateral pressure coefficient, the depth of the plastic zone of the custom constitutive model is greater than that of the Mohr–Coulomb model, and the morphology of the plastic zone is similar between the two.

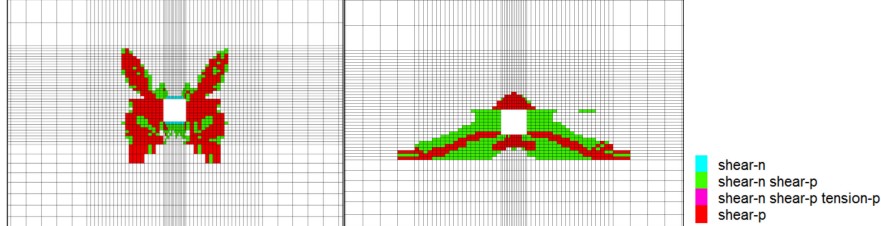

Lateral pressure coefficient 0.333; lateral pressure coefficient 6.7.

(**a**)

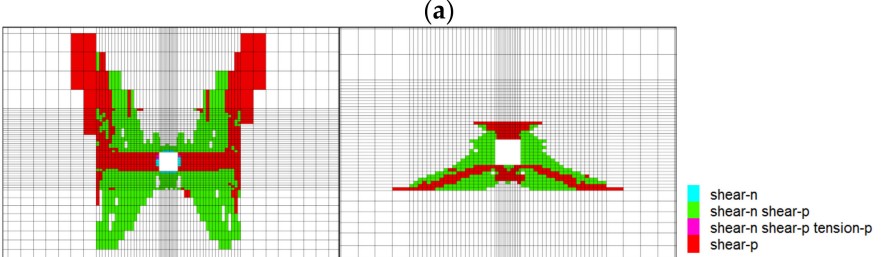

Lateral pressure coefficient 0.333; lateral pressure coefficient 6.7.

(**b**)

**Figure 14.** Plastic zone of the surrounding rock in the second group of roadways. (**a**) The plastic zone of the Mohr–Coulomb constitutive model roadway perimeter rock. (**b**) Customized constitutive model roadway perimeter rock plasticity zone.

*3.5. Law of Progressive Instability of Interface-Bearing Surrounding Rocks in High-Stress Roadways*

The initial vertical stress in the second group of roadways gradually decreases from 30 MPa to 1.5 MPa, and the initial horizontal stress in the first group of roadways gradually increases from 1.5 MPa to 30 MPa. The two groups of models correspond to different stress conditions of the roadway surrounding rock:

(1) For the interface-containing roadway with a burial depth of H, when affected by the overpressure, the roadway may be under stress conditions with a peak pressure higher than the original rock stress. K is the stress concentration coefficient and is the capacitance. At this time, the lateral pressure coefficient of the roadway is less than 1, and the higher the degree of stress concentration, the smaller the lateral pressure coefficient;

(2) When the horizontal stress of the surrounding rock is λ (λ >1) times the vertical stress, i.e., the lateral pressure coefficient of the roadway, for some, the burial depth is larger, but the horizontal stress of the surrounding rock is greater than the vertical stress of the roadway.

For the same burial depth of H, the roadway deformation and stress distribution in the above two stress states can be corresponded to the roadway in the first group (λ > 1) and the roadway in the second group (λ < 1).

A comparison of the stress and deformation distribution between the first and second groups shows that:

(1) For stress, the initial stress values of the two groups are the same, the directions are different and the stress concentrations in the two groups are also different. When the lateral pressure coefficient is 3, the roadway generates stress concentrations at the top and floor of the roadway, with a peak value of 42.5 MPa in the Mohr–Coulomb presentational model and 41 MPa in the custom presentational model; when the lateral pressure coefficient is 0.333, the stress concentration is generated in both sides of the roadway, with a peak value of 39.6 MPa in the Mohr–Coulomb model and 43.4 MPa in the custom model.

When the vertical stress is greater than the horizontal initial stress, the peak stress of the roadway envelope is concentrated at the two sidewalls of the roadway and is close to

the coal–rock interface position. As the horizontal initial stress increases and is greater than the vertical initial ground stress, the peak stress of the roadway envelope is transferred to the floor of the roadway, and the two sidewalls of the roadway and the top plate still represent the stress concentration area, but the concentration coefficient is smaller.

(2)  For deformation, the analysis is based on the premise that the initial stress values of the two groups of models are equal, but the loading directions are different, i.e., the initial vertical stress of the second group is equal to the initial horizontal stress of the first group of models.

As shown in Figure 15, as the initial horizontal stress in the first group and the initial vertical stress in the second group gradually increase, the deformation of the sidewall and the top and floor gradually increases. Under the self-defined constitutive model, when the initial stress is less than 10 MPa, the deformation of the sidewall part of the second group is larger than the model of the first group, and the deformation of the top plate is smaller than the model of the first group; when the initial stress is larger than 10 MPa, with the increase in the initial stress, the deformation of the top and floor of the first group is larger than the second group, and the deformation gap is larger, with the sidewall part of the second group larger than that of the first group.

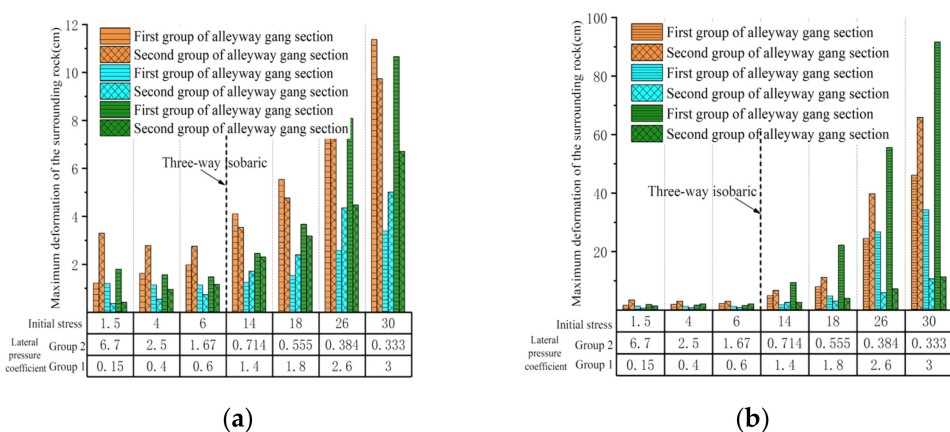

**Figure 15.** Deformation of surrounding rock at the top and floor of the roadway. (**a**) The Mohr–Coulomb model. (**b**) Customized constitutive model.

The trend of deformation calculated by the Mohr–Coulomb model and the custom constitutive model is basically the same, but the amount of deformation differs greatly. In the numerical simulation process, the values of the model parameters are selected in conjunction with the custom constitutive structure model. In the first group, when the horizontal and vertical initial stresses of the roadway are 30 MPa and 10 MPa, respectively, the floor deformation reaches 91.56 cm, while the deformation of the sidewall and top plate is 46 cm and 34.17 cm, and the floor deformation is greater. This is because the squeezing force in the horizontal direction is greater than the vertical force at this time, and the stress is concentrated in the top and floor of the roadway, so that it reaches the post-peak stage. The deformation of the roadway occurs firstly in the floor and top plates of the roadway, the existence of the interface reduces the integrity of the sidewall and the top plate of the roadway, and the deformation of the sidewall is greater than that of the top plate.

In the second group, when the horizontal and vertical initial stresses of the roadway are 10 MPa and 30 MPa, respectively, the deformation of the sidewall is the greatest, reaching 65.77 cm; the deformations of the top and floor are 10.57 cm and 11.22 cm, respectively; and the deformation of the sidewall is much larger than the deformation of the top and floor. This is because, at this time, the initial vertical stress is larger than the horizontal stress, the stress concentration zone appears at the intersection of the two sidewalls of the roadway, so that the deformation of the sidewall increases, the plastic zone of the sidewall extends to a larger extent, the stress concentration zone is far away

from the roadway, coupled with the weakening effect of interface force transfer and the deformation of the top and floor. The analysis results are consistent with the conclusions of Dubinya [39] and Liu Hongtao's [40] papers.

As shown in Figure 16, the horizontal stress is greater than the vertical stress, with the increase in the high-stress area of the roadway gradually shifting from the two sidewalls to the roof and floor of the roadway, and the stress concentration area of the floor is larger and the peak is higher. Therefore, the deformation of the floor is generally larger than that of the sidewall and the top in the roadway with high horizontal stress, so attention should be paid to the support of the floor slab. If the horizontal stress is too high, pressure relief should be applied to prevent stress concentration and further destabilization of the roadway.

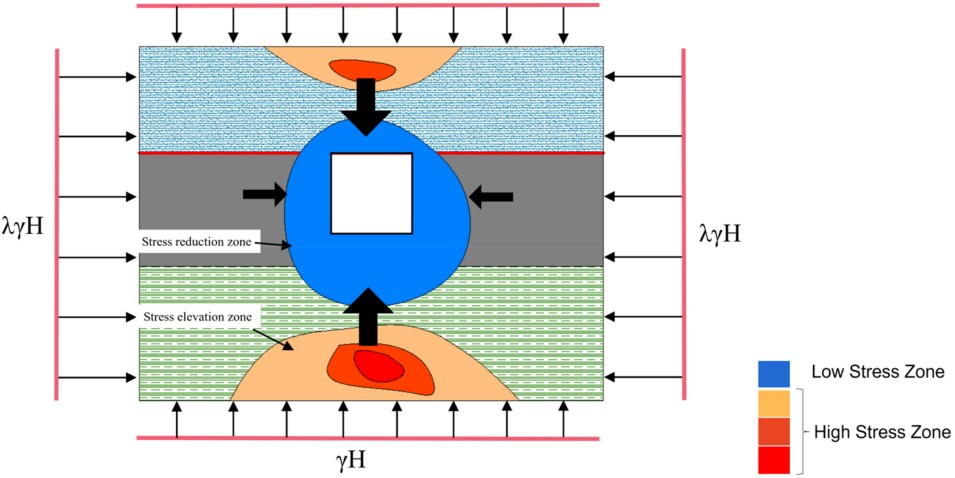

**Figure 16.** The stress and deformation distribution trend of the surrounding rock of the roadway, when the horizontal stress is greater than the vertical stress. The arrow in the figure represents the direction of stress action.

As shown in Figure 17, located in front of the working face of the mining area alley or adjacent to the working face of the recovery roadway, after recovery, the roadway is in the high-stress area. As the pressure of the top plate rises, the stress concentration coefficient K$\gamma$H value increases. At this time in the roadway fissure development, the roadway's two helpers form a stress elevation area—the stress value is greater than K, while the stress value of the top and floor is smaller; the larger the K value, the higher the stress value of the two helpers. The deformation of the top and floor is smaller than that of the two sidewalls, and the larger the K value is, the larger the deformation is, and the deformation increases exponentially.

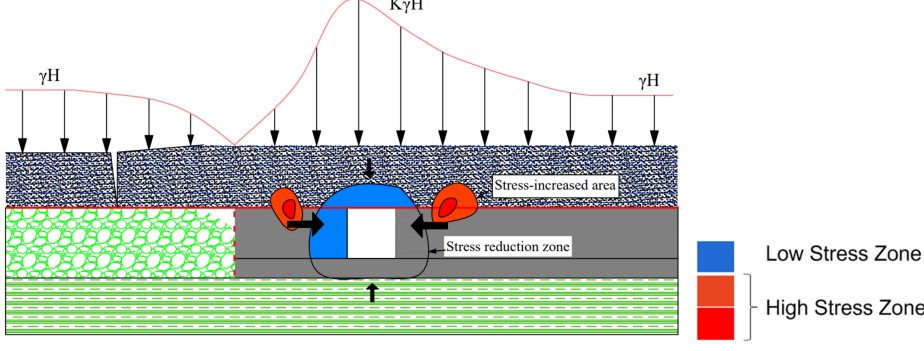

**Figure 17.** The stress and deformation distribution trend of the surrounding rock of the roadway, when the vertical stress is greater than the horizontal stress. The arrow in the figure represents the direction of stress action.

Assuming the buried depth of the roadway is 400 m, γH = 10 MPa, when K = 1, the maximum deformation of the sidewall, roof and floor are 2.9 cm, 2.1 cm and 2.1 cm. When K = 3, the deformation of the sidewall increases to 65.77 cm, and the maximum deformation of the roof and floor is 11.22 cm. The expansion of the plastic zone in the surrounding rock of the tunnel also increases sharply with the increase in K value. When K = 1, the plastic zones in the sidewall, roof and floor are 1.5 m, 0.35 m and 1.875 m. Correspondingly, When K = 3, the plastic zone is distributed in a "butterfly" shape, with the plastic zone being 12 m, 1.4 m and 3 m, the roof angle extending to 11 m, and the floor angle extending to 14 m. The free surface of the tunnel is subjected to tensile shear failure.

Therefore, support should be strengthened for the sidewall of such a roadway to prevent excessive deformation of the sidewall. Considering that the deformation of the roof and floor of the roadway is much smaller than that of the sidewall, it can be considered to strengthen the connection between the surrounding rock and the roof and floor of the tunnel sidewall within a controllable range, reducing the sliding and dislocation between the tunnel side and the rock layer. At the same time, the roadway is prevented from being in a stress environment with a high-stress concentration, and reducing the K value can significantly improve the stress state of the roadway, reduce the deformation of the roadway, and expand the plastic zone.

## 4. Conclusions

This paper uses numerical simulation for secondary development and applications through the use of a Mohr–Coulomb model, custom constitutive model comparison, and the verification of certain working conditions in the field to analyze the influence of different stress environments on the stability of the rock surrounding a high-stress-bearing laminated roadway. The main conclusions are as follows:

(1) The three-dimensional difference form of the damage body constitutive model was derived, and the constitutive equation was developed twice in FLAC3D software. The simulation results of the custom constitutive model fit well with the experimental data and are better than those of the Mohr–Coulomb constitutive model.

(2) With the change in the lateral pressure coefficient of the surrounding rock, the trend of deformation calculated by the Mohr–Coulomb model and custom constitutive model is basically the same, but the difference in the deformation amount is significant. The deformation of the Mohr–Coulomb constitutive model is significantly smaller than that of the custom constitutive model. The maximum displacement of surrounding rock follows an exponential trend with the variation in a lateral pressure coefficient.

(3) When the difference between vertical pressure and horizontal pressure increases, the plastic zone of the tunnel develops in a "butterfly" shape, and the plastic zone range of the custom constitutive model is larger than that of the Mohr Coulomb model.

(4) For roadways with high horizontal stress, the deformation of the roadway floor is generally greater than that of the sidewall and roof. Attention should be paid to the support of the floor. For roadways with high vertical pressure, support should be strengthened to prevent excessive deformation of the sidewall, reduce the sliding and dislocation between the sidewall and the rock layer, and prevent the roadway from being in a high-stress concentration environment. Reducing the K value can improve the deformation development trend of the tunnel.

**Author Contributions:** T.W. and J.C. conceived and established the numerical model. T.W. and H.W. wrote the paper. All authors have read and agreed to the published version of the manuscript.

**Funding:** This research was funded by Scientific Research Foundation for High-Level Talents of Anhui University of Science and Technology (grant number 13210673), Natural Science Research Project of Anhui Educational Committee (grant number KJ2021A0453), National Natural Science Foundation of China (grant number 52204081, 52174105).

**Institutional Review Board Statement:** Not applicable.

**Informed Consent Statement:** Not applicable.

**Data Availability Statement:** Not applicable.

**Conflicts of Interest:** The authors declare no conflict of interest.

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
