# Peer review of "A Study of the Deformation Law of the Surrounding Rock of a Laminated Roadway Based on FLAC3D Secondary Development"

_applsci, doi:10.3390/app13106077_

Round 1
Reviewer 1 Report
Reconsider after major revision.

Author Response
Dear editor,
We are sorry for the mistakes. In accordance with the suggestions of the reviewers, the paper has been revised in detail. At the same time, I am still very grateful for the suggestions of reviewers. We have sought MDPI English editing service to help us correct grammar errors.
Thank you very much.
- page 2 line 46,english check required.
Thank you very much for your patience and meticulous review.
We are sorry for the mistakes.
We have sought MDPI English editing services to help correct grammar errors from the full text.
We would like to answer the second, third, and eighth questions together, considering that these questions require similar answers.
- page 7 line 242,why the H-B model is not selected....H-B model incorporate more real rock parameters as compared to M-C model.
- page 7 line 255,include the H-B model and then compare the results
Thank you very much. We have fixed the case error of the keyword
- page 19 line 606, custom intrinsic model should be double checked with other well known models like Hoek and Brown model which are more closer to rock mass reality ....only comparison with M-C model is not sufficient
Thank you very much.
You are an expert in the field of rock mechanics, and I very much agree with your that the H-B criterion corporate more real rock parameters. There are our brief understanding of why we did not use the H-B model but instead used the M-C model
1)M-C criterion is the most widely used strength criterion in the field of rock mechanics, and its application is relatively simple.
2)In the experimental section of the article, we used the generalized H-B criterion and M-C criterion to fit the experimental data of coal rock combinations and individual coal and rock samples. We found that the two strength criteria had similar fitting results for the strength of coal rock interface rocks. The specific fitting results are as follows:
Fitting curve of generalized H-B criterion ( means the height ratio of different coal and rock)
Mohr-Coulomb strength criterion fitting curve
Fitting curves of H-B strength criterion corresponding to different m values (Experimental data for a coal rock height ratio of 0.67 )
From the fitting results, both the Mohr Coulomb criterion and the generalized Hoek Brown criterion have good fitting effects on the experimental results, with R2 reaching above 0.9.
The Mohr Coulomb criterion has a simple and practical formula, a small number of parameters, a clear physical meaning of the parameters, and each parameter value can be measured through conventional experimental methods
The parameters of the Hoek Brown criterion are greatly influenced by subjective factors.
Of course, this is only a fitting of strength, and the fitting of the stress-strain constitutive relationship also requires the establishment of a nonlinear damage model based on the m-c strength criterion.
3)The process of rock failure is actually the development and evolution of internal cracks and microcracks in the rock. In the manuscript, the strain softening characteristics of coal rock combination in the loading process are described with the theory of Damage mechanics, and the damage evolution model of coal rock combination is deduced. The interior of the rock is composed of micro elements that can bear loads, and the damage micro elements are randomly distributed, For the damaged part, scholars use different strength criteria such as Drucker-Prager criterion[1, 2] and Mogi-Coulomb criterion[3] to improve the characterization method of micro element strength.
And this manuscript proposes a rock microelement strength measurement method that can consider damage thresholds based on the Mohr-Coulomb criterion. Based on this, a nonlinear model was established and the secondary development of the model was completed, which was applied to the calculation process of tunnel deformation. This is one of the reasons why we did not use the H-B constitutive model but the Mohr Coulomb constitutive model to compare with experimental data.
[1] Cao Wengui, Zhao Heng, Zhang Yongjie, et al A constitutive model and parameter determination method for rock strain softening and hardening damage considering the influence of volume changes Geotechnical Mechanics, 2011,32 (03): 647-654.
[2] Zhang Chao Research on deformation mechanism of brittle rock and its deformation process simulation simulation method Hunan University, 2017.
[3] Jiang Bangyou, Tan Yunliang, Wang Lianguo, Gu Shitan, Dai Huabin An elastic-plastic damage constitutive model based on the Mogi Coulomb criterion and its numerical implementation Journal of China University of Mining and Technology, 2019, 48 (04): 784-792.
4)The M-C strength criterion involves fewer parameters and is relatively simple in theoretical calculations. The derivation of the three-dimensional difference form of the model is also simpler.
5)The secondary development language of the M-C constitutive model is relatively simple, involving relatively few parameters. Of course, the final simulation results and experimental data fit well, which is also one of the reasons why we use M-C constitutive model.
6)Dear editor, thank you very much for your valuable suggestions, which have also pointed out a very good research direction for our future research. We will definitely use the H-B criterion to derive the stress-strain relationship of composite materials in the future, and try to use the platform to rewrite and modify the H-B criterion to make it more in line with experimental data and actual on-site conditions.
Thank you again for your suggestion.
- page 12 line 382,why this deformation accelerates and difference become too much ....?
Thank you very much.
The lateral pressure coefficients of the model are divided into two groups. In the first group, when the vertical stress is 10MPa and the horizontal stress is 30MPa, the lateral pressure coefficient of surrounding rock is 3, and the deformation is the largest. In the second group, when the vertical stress is 30MPa and the horizontal stress is 10MPa, the lateral pressure coefficient of surrounding rock is 0.33, and the deformation is the largest. It can be seen that the magnitude of the deformation of the surrounding rock is not only related to the pressure coefficient of the surrounding rock, but more importantly, to the magnitude of the stress in the surrounding rock. This is also consistent with reality. The purpose of setting two groups is to avoid confusion caused by the difference between the stress size and the lateral pressure coefficient of surrounding rock.
When the lateral pressure coefficient of the surrounding rock is relatively small and approaches 1, the deformation is not significant. There are two reasons for the rapid increase in deformation of the surrounding rock: high stress level and the deviatoric stress caused by the lateral pressure coefficient. When rewriting the three-dimensional differential form, the custom constitutive model divides the stress of the rock into spherical stress and deviatoric stress. The spherical stress tensor only causes volume changes of the deformed object and does not cause shape changes, which does not contribute to the plastic deformation of the rock. The deviatoric stress tensor causes a change in the volume of an object. Within time step , the relationship between the stress increment and strain increment of the damaged body can be rewritten as:
Based on experimental data (published in the paper [1]), in the coal rock combination, when the peak value is reached, the axial deformation shows a sharp increase trend, and the deformation speed of the Mohr Coulomb constitutive model also increases after the peak value, but the amplitude is smaller than that of the custom constitutive model. Especially when the two conditions of high stress level and deviation stress difference caused by lateral pressure coefficient are met, the deformation difference is large.
(Cited from reference 1) Figure 4 of this article
[1] WANG T, MA Z, GONG P, et al. Analysis of failure characteristics and strength criterion of Coal-Rock combined body with different height ratios[J]. Advances in Civil Engineering, 2020, 2020: 1-14.
- page 15 line 441,increase in plastic zone is due to increase in deviatoric stress...??
Thank you very much.
There are two reasons for the rapid increase in deformation of surrounding rock: high stress levels and differential stress caused by lateral pressure coefficient. We set up two sets of experimental data to observe the deformation under two conditions.
- page 15 line 451,this butterfly damage is amazing ....double check your boundary conditions ....
Thank you very much for your patience and meticulous review.
The code for boundary conditions is as follows:
fix x y z range z -0.1 0.1
fix x range x -0.1 0.1
fix x range x 82.9 83.1
fix y range y -0.1 0.1
fix y range y 14.9 15.1
In addition, all models are conducted under the same boundary conditions, and the plastic region is similar to a butterfly shape only when the difference between vertical and horizontal stresses is significant. However, due to the limitation of space, the plastic zone under the lateral pressure coefficient of all surrounding rocks is not shown in the article.
|
|
|||
|
0.333 |
0.714 |
1.67 |
6.7 |
- page 15 line 463, Large portion around the tunnel should be in tension which is not visible from your results ....
What you said is completely correct. A part of the surrounding area of the tunnel should be in tension. The plastic zone in the figure is too large, causing some areas in the tension state to be unclear. The following image is an enlarged version of one of the figures, where some of the tension areas can be seen.
In fact, under the dual influence of the surrounding rock interface and the surrounding rock pressure measurement coefficient, the tension area around the tunnel is also somewhat different. The connection between the roof and the side wall is reduced, and the deformation of the tunnel side wall towards the inside of the tunnel is accelerated, resulting in faster pressure relief, which has a significant impact on the tension area.
In addition, we have made extensive revisions to the article, including adjusting the abstract section to the appropriate word count, enriching the citation direction in the introduction, conclusion section, reference section, etc. Thank you again for your suggestions and review.

Reviewer 2 Report
Manuscript ID: applsci-2357644
Title: Study on the deformation law of the surrounding rock of laminated roadway based on FLAC3D secondary development
by Tuo Wang, Jucai Chang and Hongda Wang
Written evaluation
Overall impression and recommendation
The manuscript investigates the impact of different stress environments on the deformation of the surrounding rocks of laminated roadways. The topic is covered in detail, but the writing style needs to be changed in places. In general, there is too much text that is harder to follow due to shortcomings which need to be addressed as stated in comments bellow.
1) language editing required - I noticed grammatical errors in the paper, such as the word order, use of articles, singular and plural, unnecessary dots, and capital letters. I, therefore, recommend that authors double-check their grammar, ask their native English-speaking colleague to assist to check grammar, and/or use the English editing service to ensure that it is meet the standard of publication. Some changes will be suggested in further comments.
2) Limit the abstract on around 200 words, as proposed by the manuscript template/instructions for authors. In the present form, there are more than twice the prescribed words.
3) Keywords: check capital letters
4) Lines 41-47: The first sentence in the Introduction extends through 7 rows, it seems that the text is scattered, it needs to be separated into 3-4 coherent sentences.
5) Lines 57-58: Please add which are these various methods.
6) Lines 98-101: sentence repetition
7) It is necessary to fix referring to the authors throughout the entire text, e.g. line 65: Quansheng et al. [16]; line 67: Zhang & Wong [17]; line 70: Chen et al. [18-19]; line 111: Wang et al. [29-30]; in the submitted version, the references are wrong
8) Figures: please increase font in all figures, where possible
9) Table 4: it seems that it needs to be Constant horizontal force/Vertical force change. Please revise
10) Section 3.4. Text missing, the section starts with figure, please revise
11) Section 4 in not actually Discussion, rather it is Summary and conclusion
12) References should be standardized according to the default format, the current version is not correct:
References
References must be numbered in order of appearance in the text (including citations in tables and legends) and listed individually at the end of the manuscript. We recommend preparing the references with a bibliography software package, such as EndNote, ReferenceManager or Zotero to avoid typing mistakes and duplicated references. Include the digital object identifier (DOI) for all references where available.
Citations and references in the Supplementary Materials are permitted provided that they also appear in the reference list here.
In the text, reference numbers should be placed in square brackets [ ] and placed before the punctuation; for example [1], [1–3] or [1,3]. For embedded citations in the text with pagination, use both parentheses and brackets to indicate the reference number and page numbers; for example [5] (p. 10), or [6] (pp. 101–105).
1. Author 1, A.B.; Author 2, C.D. Title of the article. Abbreviated Journal Name Year, Volume, page range.
2. Author 1, A.; Author 2, B. Title of the chapter. In Book Title, 2nd ed.; Editor 1, A., Editor 2, B., Eds.; Publisher: Publisher Location, Country, 2007; Volume 3, pp. 154–196.
3. Author 1, A.; Author 2, B. Book Title, 3rd ed.; Publisher: Publisher Location, Country, 2008; pp. 154–196.
4. Author 1, A.B.; Author 2, C. Title of Unpublished Work. Abbreviated Journal Name year, phrase indicating stage of publication (submitted; accepted; in press).
5. Author 1, A.B. (University, City, State, Country); Author 2, C. (Institute, City, State, Country). Personal communication, 2012.
6. Author 1, A.B.; Author 2, C.D.; Author 3, E.F. Title of Presentation. In Proceedings of the Name of the Conference, Location of Conference, Country, Date of Conference (Day Month Year).
7. Author 1, A.B. Title of Thesis. Level of Thesis, Degree-Granting University, Location of University, Date of Completion.
8. Title of Site. Available online: URL (accessed on Day Month Year).
(copied from the word template)
1) language editing required - I noticed grammatical errors in the paper, such as the word order, use of articles, singular and plural, unnecessary dots, and capital letters. I, therefore, recommend that authors double-check their grammar, ask their native English-speaking colleague to assist to check grammar, and/or use the English editing service to ensure that it is meet the standard of publication. Some changes will be suggested in further comments.
Author Response
Dear editor,
We are sorry for the mistakes. In accordance with the suggestions of the reviewers, the paper has been revised in detail. At the same time, I am still very grateful for the suggestions of reviewers. We have sought MDPI English editing service to help us correct grammar errors.
Thank you very much.
- language editing required - I noticed grammatical errors in the paper, such as the word order, use of articles, singular and plural, unnecessary dots, and capital letters. I, therefore, recommend that authors double-check their grammar, ask their native English-speaking colleague to assist to check grammar, and/or use the English editing service to ensure that it is meet the standard of publication. Some changes will be suggested in further comments.
Thank you very much for your patience and meticulous review.
We are sorry for the mistakes. We will seek MDPI English editing service to help us correct grammar errors.
- Limit the abstract on around 200 words, as proposed by the manuscript template/instructions for authors. In the present form, there are more than twice the prescribed words.
Thank you very much. We have limited the abstract to 200 words, as proposed by the manuscript template/instructions for authors.
In the original manuscript, change the following: To investigate and analyze the influence of different stress environments on the deformation and destabilization of the rocks surrounding laminated roadways under high stress, this study conducted numerical simulations of coal-rock assemblages under different circumferential pressures and of the surrounding rocks of highly stressed laminated roadways under different lateral pressure coefficients. In addition, a new custom intrinsic structure model was constructed based on the Mohr–Coulomb criterion and realized in FLAC3D software by combining field working conditions. The model was then developed in FLAC3D software for a second time. The results show that the calculated results of the model in this study are in good agreement with the experimental results and the errors are small, while the calculated results of the Mohr–Coulomb model differ from the experimental values under two types of surrounding rock pressure. The deformation of the Mohr–Coulomb model is significantly smaller than that of the customized model, which verifies the reasonableness and superiority of the self-built model in combination with the field conditions. This provides theoretical and practical bases for the design and optimization of stratigraphic roadway support in underground coal mines.
Thank you again for your suggestion.
- Keywords: check capital letters
Thank you very much. We have fixed the case error of the keyword
In the original manuscript, change the following:Keywords: Stress environment; Laminated roadway envelope; Mohr-coulomb criterion; Flac3d numerical simulation secondary development; Damage intrinsic model
- Lines 41-47: The first sentence in the Introduction extends through 7 rows, it seems that the text is scattered, it needs to be separated into 3-4 coherent sentences.
Thank you very much. We have modified the original first sentence of the introductory section (lines 41-47) into four coherent sentences (lines 26-32).
In the original manuscript, change the following:In recent years, with the increasing depth and intensity of coal mining. The high stress laminated roadway envelope presented by the coal mining gradually develops to the deeper part, which makes the original roadway support scheme based on the complete rock formation inadequate [1-5]. Therefore, there is an urgent need to explore a new support theory suitable for laminated roadway envelopes in deep high stress conditions. Firstly, it is essential to study the influence of different stress environments on the support of deep ground engineering roadway rock enclosures containing laminated roadways.
- Lines 57-58: Please add which are these various methods.
Thank you very much. In lines 57-58, we have added a different approach regarding the experiment.
In the original manuscript, change the following:And the mechanical test of coal rock body can not only provide the basic mechanical parameters of the rock, but also has an important reference significance to reveal the fracture evolution of the roadway enclosure and the failure mechanism of the enclosure . Many scholars have done mechanical property tests on coal and rock bodies, including conventional triaxial tests on pure coal and pure rock bodies, true triaxial tests and mechanical property tests on rocks under different stress paths; some experts and scholars have also conducted mechanical tests on combined rock bodies in the form of coal-rock combinations, revealing the strength characteristics and damage forms of coal-rock combinations. [12].
- Lines 98-101: sentence repetition
Thank you very much. The original repeated sentences (lines 98-101) have been removed from the repeated parts (lines 86-88).
In the original manuscript, change the following:Therefore, the study of deformation, strength characteristics and failure characteristics of a single coal or rock body cannot accurately reveal the force characteristics of such roadway, nor can it directly indicate the cause of uneven large deformation in the roadway.
- It is necessary to fix referring to the authors throughout the entire text, e.g. line 65: Quansheng et al.[16];line 67: Zhang & Wong [17]; line 70: Chen et al. [18-19]; line 111: Wang et al. [29-30]; in the submitted version, the references are wrong.
Thanks for your suggestion.
Errors and issues have been corrected in the manuscript.
- Figures: please increase font in all figures, where possible.
Thanks for your suggestion.
Thanks for your suggestion.
Font in figures have been corrected in the manuscript.
- Table 4: it seems that it needs to be Constant horizontal force/Vertical force change. Please revise
Thank you very much. We have corrected the errors in Table 4.
In the original manuscript, change the following:
|
Lateral pressure coefficient |
0.333 |
0.384 |
0.555 |
0.714 |
1 |
1.67 |
2.5 |
6.7 |
|
Constant horizontal stress /MPa |
10 |
|||||||
|
Vertical stress change /MPa |
30 |
26 |
18 |
14 |
10 |
6 |
4 |
1.5 |
- Section 3.4. Text missing, the section starts with figure, please revise.
Thank you very much. The description statement has been added.
Comparison and extraction of the distribution pattern of plastic zone in the roadway surrounding rock under different lateral pressure coefficients using Mohr-Coulomb and custom constitutive model.
- Section 4 in not actually Discussion, rather it is Summary and conclusion.
Thanks for your suggestion.
The title of section 4 has been corrected.
- References should be standardized according to the default format, the current version is not correct.
Thank you very much. We have revised the references according to the canonical format.
In the original manuscript, change the following:
- Xie, H.P.; Gao, F.; Ju, Y.; Gao, M.Z.; Zhang, R.; Gao, Y.A.; Liu, J.F.; Xie, L.Z. Quantitative definition and investigation of deep mining .Journal of China Coal Society, (01):1-10. DOI:10.13225/j.cnki.jccs.2014.1690.
- Xie, H.P.; Gao, F.; Ju, Y. Research and development of rock mechanics in deep ground engineering . Chinese Journal of Rock Mechanics and Eegieneering,2015,34(11):2161-2178. DOI:10.13722/j.cnki.jrme.2015.1369.
- Hou, C.J.; Wang, X.Y.; Bo, J.B.; Meng, N.K.; Wu, W.D. Basic theory and technology study of stability control for surrounding rock in deep roadway. Journal of China University of Mining & Teachnology,2021,50(01):1-12. DOI:10.13247/j.cnki.jcumt.001242.
- Kang, H.P. Seventy years development and prospects of strata control technologies for coal mine roadway in China . Chinese Journal of Rock Mechanics and Eegieneering,2021,40(01):1-30. DOI:10.13722/j.cnki.jrme.2020.0072.
- Kang, H.P.; Wang, G.F.; Jiang, P.F.; Wang, J.C.; Zhang, N.; Jing, H.W.; Huang, B.X.; Yang, G.; Guan, X.M.; Wang, Z.G. Conception for strata control and intelligent mining technology in deep coal mines with depth more than 1 000 m .Journal of China Coal Society,2018,43(07):1789-1800. DOI:10.13225/j.cnki.jccs.2018.0634.
- Zhan, G.W.; Xia, Y.C.; Du, R.J. Application of Hoek-Brown Strength Criterion in Numerical Simulation of FLAC3D .Journal of Mining 8 Safety Engineering,2007,82(03):366-369.
- Ren, Y.L.; Yang, W.H.; Han, T. Study of the ultimate bearing capacity of a shaft lining based on different failure criteria . Journal of China University of Mining & Teachnology,2011,40(04):540-543.
- Hu, S.R.; Yu, M.H. Three-shear strength criterion and its application to the elastic-plastic analysis of roadway envelopes. Journal of China Coal Society,2003,(04):389-393.
- Ding, X.; Xiao, X.C.; Pan, Y.S.; Lv, X.F.; Wu, D. Coal Constitutive Relation and Mechanical Analysis of Its Impact Tendency Index . Chinese Journal of Underground Space and Engineering,2020,16(05):1371-1382.
- Peng, R.; Meng, X.R.; Zhao, G.M.; Li, Y.M.; Zuo, C. Surrounding rock unified solution near deep-buried roadway excavation work face considering incremental constitutive relationship. Journal of China University of Mining & Teachnology,2015,44(03):444-452. DOI:10.13247/j.cnki.jcumt.000328.
- Li, Q.W.; Gao, F.L.; Hu, L.L.; Yu M.M.; Liu, Y.W.; Zeng, X.G.; Zhu, Q.Y.; Cao, X.; Huang, X. Constitutive relation of energy dissipation damage of heterogeneous coal samples under different loading rates.Journal of China Coal Society,2022, 47(S1):90-102. DOI:10.13225/j.cnki.jccs.2022.0163.
- Wang, T. Research and Application of Deformation Mechanism of Surrounding Rock with Bedding Surface in High Stress Roadway [D]. China University of Mining and Technology,2021. DOI:10.27623/d.cnki.gzkyu.2021.000022.
- Hu, X.L.; Qu, S.J.; Li, K.Q. Study of rock elastoplastic constitutive damage model based on the unified strength theory . Journal of China University of Mining & Teachnology,2019,48(02):305-312. DOI:10.13247/j.cnki.jcumt.000985.
- Zhang, C.Q.; Zhou, H.; Feng, X.T. Numerical format of elastoplastic constitutive model based on the unified strength theory in FLAC3D . Rock and Soil Mechanics,2008,147(03):596-602. DOI:10.16285/j.rsm.2008.03.038.
- Hu, X.L.; Li, K.Q.; Qu, S.J. Rock elastic-plastic intrinsic model based on unified strength theory and its numerical implementation. Explosion and Shock Waves,2019,39(08):130-138.
- Liu, Q.S.; Liu, K.D.; Zhu, J.B. et al. Study of mechanical properties of raw coal under high stress with triaxial compression. Chinese Journal of Rock Mechanics and Eegieneering,2014,33(01):24-34. DOI:10.13722/j.cnki.jrme.2014.01.003.
- Zhang, X.P.; Wong, L.N.Y. Cracking Processes in Rock-Like Material Containing a Single Flaw Under Uniaxial Compression: A Numerical Study Based on Parallel Bonded-Particle Model Approach.Rock Mechanics and Rock Engineering,2012,45(5):711-737.
- Chen, Z.F.; Lin, Z.M.; Xie, H.H.; Wang, H.W. Damage study on brittle rock failure under complicated stress .Journal of China Coal Society,2004,(01):31-35.
- Chen, Z.H.; Tan, G.H.; Yang, W.Z. Numerical simulation of damage and failure of rocks under different confining pressures . Chinese Journal of Rock Mechanics and Engineering,2001,(05):576-580.
- Song, W.D.; Ming, S.X.; Wang, X.; Du, J,H. Experimental study of rock compression-damage-failure process . Chinese Journal of Rock Mechanics and Eegieneering,2010,29(S2):4180-4187.
- Liu, Y.K.; Long, Z.X.; Deng, Q.; Jiang, M.J.; Qiu, X.J.; Wang, F.C.; Wang, H.T.; Niu, Y. Energy evolution law and failure characteristics during coal loading and damaging process . Journal of XI'AN University of Science and Technology,2023,43(01):65-72. DOI:10.13800/j.cnki.xakjdxxb.2023.0108.
- Zhou, B.; Xu, J.; Peng, S.J.; Zhao, P.X.; Qin, L.; Bai, Y.; Cheng, L. Evolution of mechanical state and failure tendency of loaded outburst coal .Journal of China Coal Society,2022,47(03):1260-1274. DOI:10.13225/j.cnki.jccs.xr21.1711.
- Wen, J.; Tang, Z.; Dyson, A.P. The mechanical behavior of pre-existing transverse cracks in lignite under uniaxial compression.Geomechanics and Geophysics for Geo-Energy andGeo-Resources,2021,7(1).
- Zuo, J.P.; Chen, Y.; Zhang, J.W.; Wang, J.T.; Sun, Y.J.; Jiang, G.T. Failure behavior and strength characteristics of coal-rock combined body under different confining pressures .Journal of China Coal Society,2016,41(11):2706-2713. DOI:10.13225/j.cnki.jccs.2016.0456.
- Zuo, J.P.; Song, H.Q.; Chen, Y.; Li, Y.H. Post-peak progressive failure characteristics and nonlinear model of coal-rock combined body .Journal of China Coal Society,2018,43(12):3265-3272. DOI:10.13225/j.cnki.jccs.2018.0292.
- Zuo, J.P.; Xie, H.P.; Wu, A.M.; Liu, J.F. Investigation on failure mechanisms and mechanical behaviors of deep coal-rock single body and combined body . Chinese Journal of Rock Mechanics and Eegieneering,2011,30(01):84-92.
- Yang, K.; Liu, W.J.; Dou, L.T.; Chi, X.L.; Wei, Z.; Fu, Q. Experimental investigation into interface effect and progressive instability of coal-rock combined specimen .Journal of China Coal Society,2020,45(05):1691-1700. DOI:10.13225/j.cnki.jccs.DY20.0294.
- Guo, X.F.; Guo, L.F.; Ma, N.J.; Zhao, Z.Q.; Li, C. Applicability analysis of the roadway butterfly failure theory . Journal of China University of Mining & Teachnology,2020,49(04):646-653+660. DOI:10.13247/j.cnki.jcumt.001169.
- Wang, W.J.; Guo, G.Y.; Zhu, Y.J.; Yu, W.J. Malignant development process of plastic zone and control technology of high stress and soft rock roadway .Journal of China Coal Society,2015,40(12):2747-2754. DOI:10.13225/j.cnki.jccs.2015.0186.
- Wang, W.J.; Yuan, C.; Yu, W.J.; Wu, H.; Peng, W.Q.; Peng, G.; Liu, X.S.; Dong, E.Y. Stability control method of surrounding rock in deep roadway with large deformation .Journal of China Coal Society,2016,41(12):2921-2931. DOI:10.13225/j.cnki.jccs.2016.1115.
- Yuan, Y.; Wang, W.J.; Yuan, C.; Yu, W.J.; Wu, H.; Peng, W.Q. Large deformation failure mechanism of surrounding rock for gate roadway under dynamic pressure in deep coal mine .Journal of China Coal Society,2016,41(12):2940-2950. DOI:10.13225/j.cnki.jccs.2016.1119.
- Li, G.C.; Yang, S.; Sun, Y.T.; Xu, J.H.; Li, J.H. Research progress of roadway surrounding strata rock control technologies under complex conditions. Coal Science and Technology,2022,50(06):29-45. DOI:10.13199/j.cnki.cst.2022-0304.
- Pei, M.S.; Lu, Y.; Guo, W.B.; Wang, F.L.; Zhao, Z.Q. The research on stability and supporting technology of rock in gob-side entry in thick seam with parting .Journal of Mining 8 Safety Engineering,2014,31(06):950-956. DOI:10.13545/j.issn1673-3363.2014.06.020.
- Dai, B.B.; Li, H.B.; Zhang, S.J.; Zhao, X.D. Numerical Simulation on Stability of Surrounding Rock in Deep Hard Rook Tunnel Induced by Mining . Metal Mining,2021,No.538(04):70-75. DOI:10.19614/j.cnki.jsks.202104010.

Reviewer 3 Report
The manuscript: Study on the deformation law of the surrounding rock of laminated roadway based on FLAC3D secondary development
presents and discuss some original models or rather computer simulation concerning the deformation of roadway bases.
From the scientific point of view, the manuscript presents interest for readers, but in its actual form, it can not be published.
Excepting some remarks which could be found on the attached annotated pdf. file, the main weakness regards its length. The abstract has 438 words,
which is about two times and half the usual length of an abstract. Besides it, the entire text should be reviewed by a native speaking English.
22 pages are also, in my opinion too much for a paper that presents the results of computer simulations.
In my opinion, 12 -14 pages would be enough.
For this reason, I do not recommend the manuscript to be published in its present version, but after a thorough revision.

Author Response
Dear editor,
We are sorry for the mistakes. In accordance with the suggestions of the reviewers, the paper has been revised in detail. At the same time, I am still very grateful for the suggestions of reviewers.
Thank you very much for your meticulous and patient review. Our main purpose is to explain what research we have conducted and how it has been helpful to the site. Therefore, it has taken up a considerable amount of space. We have made possible deletions to the content while ensuring that the original meaning remains unchanged. Thank you for your support and understanding.
We have sought MDPI English editing service to help us correct grammar errors.
Thank you very much.
- The Abstract is too long. Please restrain it to maximum 200 words.
Thank you very much for your patience and meticulous review.
We have limited the abstract to 200 words, as proposed by the manuscript template/instructions for authors.
In the original manuscript, change the following:To investigate and analyze the influence of different stress environments on the deformation and destabilization of the rocks surrounding laminated roadways under high stress, this study conducted numerical simulations of coal-rock assemblages under different circumferential pressures and of the surrounding rocks of highly stressed laminated roadways under different lateral pressure coefficients. In addition, a new custom intrinsic structure model was constructed based on the Mohr–Coulomb criterion and realized in FLAC3D software by combining field working conditions. The model was then developed in FLAC3D software for a second time. The results show that the calculated results of the model in this study are in good agreement with the experimental results and the errors are small, while the calculated results of the Mohr–Coulomb model differ from the experimental values under two types of surrounding rock pressure. The deformation of the Mohr–Coulomb model is significantly smaller than that of the customized model, which verifies the reasonableness and superiority of the self-built model in combination with the field conditions. This provides theoretical and practical bases for the design and optimization of stratigraphic roadway support in underground coal mines.
Thank you again for your suggestion.
- Too long phase. Please reformulate.
Thanks for your suggestion.
Original phase:
In actual geological engineering, there are a large number of fractured rocks, and the primary fractures and voids inside the rocks are closed during the initial loading, when the volume of the rocks becomes smaller and shows the characteristics of volume compression.
Modified phase:
In actual geological engineering, there are a large number of fractured rocks. The original cracks and pores inside the rock are closed during the initial loading process, exhibiting a characteristic of volume compression.
We have also simplified the following sentences:
Page3 line 139. In practical engineering, due to the contact between the roof rock and the coal seam to be mined, thus forming a laminated roadway enclosure under high stress and different lateral pressure coefficients.
Page2 line 54. Meanwhile, some experts and scholars have also conducted mechanical tests and revealed the strength characteristics and damage forms of coal-rock combinations.
Page3 line 144. The innovations are as follows: the stability of high stress layered roadway surrounding rock under different pressure measurement coefficients was investigated based on the secondary development and application of FLAC3D.
- Only the family name is necessary
Thanks for your suggestion.
All citation authors' full names have been replaced with family name.
- Please explain what REPA2D means.
Thank you very much. The meaning of REPA2D is two-dimensional rock failure process. And it has been noted in the article
- please, explain the term Kplease explain the term D
Thank you very much for pointing out these errors.
The meaning of the letter K and D are indicated in line 187 page 5 of the article, but we marked the wrong position and should have marked it where the letter first appeared.
K is the bulk modulus; Dt is the damage variable.
Thanks again for your suggestion.
- please include error bars
Thanks for your suggestion.
The experimental data refers to data from my own published articles, and the citation label has been marked in the corresponding position of the manuscript.
- please use larger fonts an a minimum 900 dpi imagefor figure 1.
Thank you very much for pointing out these errors.
The image has been modified.
- Another huge phrase
Thank you very much.
The paragraph has been simplified as follows:
After loading the constitutive model, verify its calculation results and experimental results. The height and diameter of the rock and coal established by numerical simulation are both 50mm, as shown in Figure 2. The loaded surrounding rock pressures are 5MPa and 10MPa respectively, and the axial the loading velocity is 1 × 10–7 mm/step. The parameters F0, and m are based on Table 1. The physical and mechanical parameters of the rock selected for the simulation test are consistent with the test, as shown in Table 2.
- which rock? sandstone, basalt, granite,...????
The article uses sandstone, and all rocks are from the same batch of samples, which have been modified in the manuscript.
- it is very difficult to read and understand a 150 words phase. Please divide into few sentences.
Thanks for your suggestion. Long paragraphs are divided into small paragraphs, and the meaning of the expression remains unchanged.
Figures 7 and 8 show the horizontal stress distribution in the laminated roadway under two different sets of loading methods, respectively, and it can be seen that:
(1)Under the Mohr–Coulomb intrinsic model, when the lateral pressure coefficient is less than 1, the horizontal stress is concentrated at the coal–rock interface, and the peak point at the interface is 2.5~3 m away from the top corner of the roadway.
(2)As the horizontal stress rises, i.e., the lateral pressure coefficient of the surrounding rock rises, the stress concentration area gradually shifts to the floor.
(3)The custom model has a stress concentration area farther from the roadway and a larger pressure relief zone around the roadway than the Mohr–Coulomb constitutive model.
We have also simplified the following sentences:
In page 12, line 376:
(1)The maximum displacement of the tunnel surrounding rock increases exponentially with the increase in the lateral pressure coefficient;
(2)The deformation of the surrounding rock in the roadway is small, when the lateral pressure coefficient is less than 1. As the lateral pressure coefficient increases, the increase in deformation and the difference of deformation between the two constitutive models is small;
(3)When the lateral pressure coefficient is greater than 1, the growth rate of surrounding rock deformation accelerates, and the difference between the two constitutive models gradually increases;
(4)When the lateral pressure coefficient is 3, the maximum displacements of the custom constitutive model sidewall, roof and floor are 46 cm, 34.17 cm and 91.56 cm, respectively, and the corresponding maximum displacements of the Mohr–Coulomb constitutive model are 11.37 cm, 3.38 cm and 10.65 cm, respectively, with the calculation results of the two constitutive models differing greatly;
(5)When the lateral pressure coefficient is 3, the maximum displacements of the custom constitutive model sidewall, roof and floor are 22.1 times, 30.8 times and 63.1 times of the lateral pressure coefficient of 0.6, respectively, and the corresponding Mohr–Coulomb constitutive models are 5.8 times, 3.0 times and 7.3 times, respectively.
In page 15, line 447:
Under the custom constitutive model, when the lateral pressure coefficient is less than 0.4, the roadway shows "butterfly" damage, when the lateral pressure coefficient is greater than 1.0, with the increase in the lateral pressure coefficient, the expansion of the plastic zone of the roadway surrounding rock is accelerated, and the plastic zone of the roadway appears similar to "butterfly" damage, with the plastic zone of the two top corners and the two floor corners developing more than the sidewall and middle top of the roadway;
Page 19, line 581
Assuming the buried depth of the roadway is 400m, γH=10MPa, when K=1, the maximum deformation of the sidewall, roof and floor are 2.9cm, 2.1cm and 2.1cm. When K=3, the deformation of the sidewall increases to 65.77cm, and the maximum deformation of the roof and floor is 11.22cm. The expansion of the plastic zone in the surrounding rock of the tunnel also increases sharply with the increase of K value. When K=1, the plastic zone in the sidewall, roof and floor are 1.5m, 0.35m and 1.875m. Correspondingly, When K=3, the plastic zone is distributed in a "butterfly" shape, with the plastic zone being 12m, 1.4m and 3m, the roof angle extending to 11m, and the floor angle extending to 14m. The free surface of the tunnel is subjected to tensile shear failure.
Therefore, support should be strengthened for the sidewall of such roadway to prevent excessive deformation of the sidewall. Considering that the deformation of the roof and floor of the roadway is much smaller than that of the sidewall, it can be considered to strengthen the connection between the surrounding rock and the roof and floor of the tunnel sidewall within a controllable range, reducing the sliding and dislocation between the tunnel side and the rock layer; At the same time, preventing the roadway from being in a stress environment with high stress concentration, reducing the K value can significantly improve the stress state of the roadway, reduce the deformation of the roadway, and expand the plastic zone.
Page 19, line 606
(1)The three-dimensional difference form of the damage body constitutive model was derived, and the constitutive equation was developed twice in FLAC3D software. The simulation results of the custom constitutive model fit well with the experimental data and are better than those of the Mohr–Coulomb constitutive model.
(2) With the change in the lateral pressure coefficient of the surrounding rock, the trend of deformation calculated by Mohr–Coulomb model and custom constitutive model is basically the same, but the difference in deformation amount is significant. The deformation of Mohr–Coulomb constitutive model is significantly smaller than that of custom constitutive model; The maximum displacement of surrounding rock follows an exponential trend with the variation of lateral pressure coefficient
(3)When the difference between vertical pressure and horizontal pressure increases, the plastic zone of the tunnel develops in a "butterfly" shape, and the plastic zone range of the custom constitutive model is larger than that of the Mohr Coulomb model;
(4)For roadway with high horizontal stress, the deformation of the roadway floor is generally greater than that of the sidewall and roof. Attention should be paid to the support of the floor. For roadway with high vertical pressure, support should be strengthened to prevent excessive deformation of the sidewall, reduce the sliding and dislocation between the sidewall and the rock layer, and prevent the roadway from being in a high stress concentration environment. Reducing the K value can improve the deformation development trend of the tunnel.
And we delete the sentence in page 10 line 304-306.

Reviewer 4 Report
Ensuring the stability of underground workings in the conditions of stratified rock mass and different stress values requires appropriate modeling reflecting industrial conditions. The presented zones of increased stress values around excavations together with their identification and location may be helpful both at the stage of excavation and strengthening of excavations. Below are some comments and suggestions:
1. Line 74, repeated word "damage";
2. In the introduction, some information should be added regarding the modeling of the joint at the coal-surrounding rock contact, in particular, please refer to the area of reduced cohesiveness at this contact;
3. In the subsection 2.1, it should be written whether the FISH programming language was used in the algorithm;
4. Table 1, it should be corrected Mpa to MPa. In addition, the full name for F0, m and R2 must be added; 5.
For the drawing, it should be explained (provide possible reasons) why the fact that at a strain value of 0.01 the black line increases and the other two lines: pink and blue decrease;
6. In the subsection 3.1, information should be added whether the presented values of vertical and horizontal stresses in Figures 5 and 6 correspond to the value in real conditions for the considered area of the mine;
7. In the subsection 3.2, it should be added information about the dimensions of the disc for figures 7-10. In addition, a spatial view of the model made in Flac3D would be useful - it would significantly enrich the value of the article;
8. Line 375, sentence: “As you can see in the figure” – should be replaced;
9. In the subsection 3.3, two explanatory sentences should be added: Gang Ministry, Roof and Base plate;
10. In Figures 13 and 14, mark the maximum vertical and horizontal plastic range in meters;
11. Figures 16 and 17 write whether the existing zones are model or actual values;
12. In the fourth chapter concerning the discussion, reference should be made to several literature items in which the influence of stratification of the rock mass on the state of stress around the excavations was examined - so that they could be compared to the results;
13. A chapter with conclusions would be useful in the article, which would largely refer to numerical modeling and recommendations for mining plants regarding the protection of the excavation under conditions of high horizontal stresses and stratification of the rock mass.
Author Response
Dear editor,
We are sorry for the mistakes. In accordance with the suggestions of the reviewers, the paper has been revised in detail. At the same time, I am still very grateful for the suggestions of reviewers. We have sought MDPI English editing service to help us correct grammar errors.
Thank you very much.
- Some errors were still not corrected as you responded. Line 74, repeated word "damage".
Thank you very much for your patience and meticulous review.
We are sorry for the mistakes. Repeated words have been deleted.The expression error has been corrected in the manuscript.
In the original manuscript, change the following:and numerically simulated the damage evolution process of rocks under different peritectic pressures by using RFPA2D.
- In the introduction, some information should be added regarding the modeling of the joint at the coal-surrounding rock contact, in particular, please refer to the area of reduced cohesiveness at this contact.
Thank you very much. The content of coal rock interface modeling information has been added in the introduction.
For the modeling of coal rock interfaces, Sun Haitao [35] studied the mechanical characteristics (plastic zone, stress, and displacement) of coal-rock composite structure under different interface connection modes by numerical simulation, and analyzed the energy by FISH language in FLAC3D. Zhao Xuemei [36] proposed a universal method to achieve pre-perception and accurate recognition of coal-rock interfaces, meanwhile, a convolutional block attention module was employed to improve the coal-rock interface identification ability of the proposed network. Zuo Jianping [37] presented experimental and numerical investigations on the response of rock-coal, coal-rock, and rock-coal-rock bimaterial composite structures under triaxial compression and the corresponding numerical simulations of the experiments were carried out by using the particle flow code.
- In the subsection 2.1, it should be written whether the FISH programming language was used in the algorithm.
Thank you very much. We have specified the use of the FISH programming language in section 2.1.
In the FLAC3D main program, the FISH programming language was used to judge whether the unbalanced force converges or not, if not, return to the second step using the updated damage variables, otherwise, the program ends. The program is run and loaded as shown in Figure 1.
- Table 1, it should be corrected Mpa to MPa. In addition, the full name for F0, m and R2 must be added.
Thank you very much for pointing out these errors. The error Mpa to MPa has been fixed and the full names of F0, m and R2 have been added.
In the original manuscript, change the following:
|
Surrounding rock pressure σ |
δ |
Micro-element intensity weibull distribution parameters F0 |
Micro-element intensity weibull distribution parameters m |
Goodness of fit R2 |
|
|
0.33 |
14.7538 |
3.2466 |
0.944 |
|
5MPa |
0.5 |
13.5508 |
5.9780 |
0.906 |
|
0.67 |
14.7134 |
3.9261 |
0.946 |
|
|
|
0.75 |
12.7565 |
4.9362 |
0.961 |
|
|
0.33 |
21.9997 |
5.8555 |
0.901 |
|
10MPa |
0.5 |
25.6579 |
3.8349 |
0.893 |
|
0.67 |
27.1918 |
4.5514 |
0.875 |
|
|
|
0.75 |
20.4382 |
3.8718 |
0.853 |
- For the drawing, it should be explained (provide possible reasons) why the fact that at a strain value of 0.01 the black line increases and the other two lines: pink and blue decrease.
Thank you very much.
The black line indicates that when the Mohr Coulomb constitutive model is used, after reaching the peak strength, the stress-strain relationship of the rock exhibits an ideal elastic-plastic form that conforms to the Mohr Coulomb constitutive model, that is, the strength remains unchanged and the deformation continues to increase. The blue curve shows the phenomenon of post peak strain softening in rocks under the custom constitutive model. The pink curve is the result of the experiment, and the curve changes of the customized constitutive model have good consistency with the experimental results.
- In the subsection 3.1, information should be added whether the presented values of vertical and horizontal stresses in Figures 5 and 6 correspond to the value in real conditions for the considered area of the mine.
Thanks for your suggestion. Tunnel excavation, coal seam mining, or some rock mass structures may cause changes in the lateral pressure coefficient of the surrounding rock, and the variation value may not reach 6.7 or 0.1 or less. The variation of the lateral pressure coefficient of the surrounding rock in the article directly increases from 2.5 to 6.7, and no further variation values are set in the middle for analysis, only as a limit case analysis. The expression regarding the setting of lateral pressure coefficient in the text has been modified as follows:
In the process of roadway layout, in fact, the roadway is often in a three-way unequal pressure state due to the influence of mining stress and the increases of burial depth. When excavating tunnels or mining coal seams, the horizontal stress on the goaf side sharply decreases, and the lateral pressure coefficient of the surrounding rock is relatively small. The horizontal stress variation of the non goaf side surrounding rock is relatively small. Due to the generated stress concentration and large vertical stress addition, the lateral pressure coefficient of the surrounding rock can change from 0 to 2, or even greater.
- In the subsection 3.2, it should be added information about the dimensions of the disc for figures 7-10. In addition, a spatial view of the model made in Flac3D would be useful - it would significantly enrich the value of the article.
- Line 375, sentence: “As you can see in the figure” – should be replaced.
Thank you very much. The sentence "As you can see in the figure" in line 375 has been replaced by the sentence "The figures show that " in line 374.
In the original manuscript, change the following:
The figures show that:
(1)The maximum displacement of the tunnel surrounding rock increases expo-nentially with the increase of the lateral pressure coefficient, when the lateral pressure coefficient is less than 1, the deformation of the tunnel surrounding rock is small, the increase of the lateral pressure coefficient, the increase of the surrounding rock defor-mation is small, and the difference between the Mohr-Coulomb principal model and the custom principal model is relatively small; when the lateral pressure coefficient is greater than 1, the growth rate of the surrounding rock deformation accelerates, and the difference between the two principal models gradually increases;
- In the subsection 3.3, two explanatory sentences should be added: Gang Ministry, Roof and Base plate.
Thank you very much for pointing out these errors.
In the original manuscript, Gang Registry should be the sidewall of the roadway, Roof and Base plate should be the roof and floor of the roadway. This is a specific noun in the roadway. I'm very sorry for the mistake caused.
- In Figures 13 and 14, mark the maximum vertical and horizontal plastic range in meters.
Thanks for your suggestion.
The figure in the text has been modified with plastic zone size identification as follows:
- 1Figures 16 and 17 write whether the existing zones are model or actual values.
Thank you very much. The existing zones of Figures 16 and 17 have been shown to be model values.
In the original manuscript, change the following:(1) but the amount of deformation differs greatly; In the numerical simulation process, the values of the model parameters are selected in conjunction with the custom intrinsic structure model, in the first group when the horizontal and vertical initial stresses of the roadway are 30MPa and 10MPa respectively; (2) In the numerical simulation process, the model values are selected as follow: suppose the burial depth of the roadway is 400m, γH=10MPa, when K=1, the deformation of the gang is 2.9cm.
- In the fourth chapter concerning the discussion, reference should be made to several literature items in which the influence of stratification of the rock mass on the state of stress around the excavations was examined - so that they could be compared to the results.
Thanks for your suggestion. The fourth chapter “Discussion” should be the section of Conclusion , and the title was mistakenly written for discussion and has been corrected. Compared with the papers of Dubinya [38] and Liu Hongtao's[39], the results are more consistent.
- A chapter with conclusions would be useful in the article, which would largely refer to numerical modeling and recommendations for mining plants regarding the protection of the excavation under conditions of high horizontal stresses and stratification of the rock mass.
Thanks for your suggestion. The fourth chapter “Discussion” should be the section of Conclusion , and the title was mistakenly written for discussion and has been corrected. Some modifications have been made to the conclusion section.

Round 2
Reviewer 1 Report
Paper may be accepted in its present form.
Paper may be accepted in its present form.
Author Response
Dear editor,
Thank you very much for your patience and meticulous review. I am very grateful for the suggestions of reviewers. Grammar errors have been corrected through MDPI English editing service.
Thank you very much.
Reviewer 3 Report
Excepting some small remarks, in its present form the manuscript worth publishing.

Author Response
Dear editor,
We are sorry for the mistakes. Thank you very much for your patience and meticulous review. I am very grateful for the suggestions of reviewers.
In accordance with the suggestions of the reviewers, the paper has been revised in detail.
- The space was inserted at front of “[12], [13-15], [20], [38] , [39], [40]”. That is a very meticulous work, thank you for your correction.
- In page 3, line 121
The phase was replaced by “ On the basis of Mohr–Coulomb criterion, the micro element strength measurement method proposed by Wang [38] that can consider the damage threshold, and introduces the derivation method and results of the damage model of coal in the coal–rock combination.”
Thank you very much.
